# The Role of LAIR1 as a Regulatory Receptor of Antitumor Immune Cell Responses and Tumor Cell Growth and Expansion

**DOI:** 10.3390/biom15060866

**Published:** 2025-06-13

**Authors:** Alessandro Poggi, Serena Matis, Chiara Rosa Maria Uras, Lizzia Raffaghello, Roberto Benelli, Maria Raffaella Zocchi

**Affiliations:** 1Molecular Oncology and Angiogenesis Unit, IRCCS Ospedale Policlinico San Martino, 16132 Genoa, Italy; serena.matis@hsanmartino.it (S.M.); chiararosamaria.uras@hsanmartino.it (C.R.M.U.); lizzia.raffaghello@hsanmartino.it (L.R.); roberto.benelli@hsanmartino.it (R.B.); 2Division of Immunology, Transplantation and Infectious Diseases, Ospedale San Raffaele, 20162 Milan, Italy; marazocchi55@gmail.com

**Keywords:** LAIR1, immune regulation, immunotherapy, immune checkpoint, inhibitory receptors

## Abstract

It is becoming evident that the therapeutic effect of reawakening the immune response is to limit tumor cell growth and expansion. The use of immune checkpoint inhibitors, like blocking antibodies against programmed cell death receptor (PD) 1 and/or cytotoxic T lymphocyte antigen (CTLA) 4 alone or in combination with other drugs, has led to unexpected positive results in some tumors but not all. Several other molecules inhibiting lymphocyte antitumor effector subsets have been discovered in the last 30 years. Herein, we focus on the leukocyte-associated immunoglobulin (Ig)-like receptor 1 (LAIR1/CD305). LAIR1 represents a typical immunoregulatory molecule expressed on almost all leukocytes, unlike other regulatory receptors expressed on discrete leukocyte subsets. It bears two immunoreceptor tyrosine-based inhibitory motifs (ITIMs) in the intracytoplasmic protein domain involved in the downregulation of signals mediated by activating receptors. LAIR1 binds to several ligands, such as collagen I and III, complement component 1Q, surfactant protein D, adiponectin, and repetitive interspersed families of polypeptides expressed by erythrocytes infected with Plasmodium malariae. This would suggest LAIR1 involvement in several cell-to-cell interactions and possibly in metabolic regulation. The presence of both cellular and soluble forms of LAIR would indicate a fine regulation of the immunoregulatory activity, as happens for the soluble/exosome-associated forms of PD1 and CTLA4 molecules. As a consequence, LAIR1 appears to play a role in some autoimmune diseases and the immune response against tumor cells. The finding of LAIR1 expression on hematological malignancies, but also on some solid tumors, could open a rationale for the targeting of this molecule to treat neoplasia, either alone or in combination with other therapeutic options.

## 1. Introduction

It is well-established that immune cells express surface receptors interacting with ligands, delivering either positive or negative signals that lead to the activation or inhibition of cellular functions [1,2,3,4,5,6,7,8,9]. Consequently, the final outcome of several receptor–ligand interactions can be considered a result of the types of receptors expressed on a leukocyte and the corresponding presence of ligands [2]. This already complex network of signals is regulated by the presence of co-stimulatory and co-inhibitory receptors, multiple ligands for the same receptor, and soluble and/or microvesicle-associated receptors [2,10,11,12,13,14,15,16,17]. This premise indicates that it is difficult to define the precise role of a given molecule in this intricate system, and the clinical results of the therapeutic targeting of a receptor/ligand molecule pair are greatly unpredictable [10,11,12,13,14,15,16,17]. The targeting of either receptors and/or ligands with appropriate tools such as monoclonal antibodies (mAbs) can regulate the function of a given receptor [18,19,20]. In several instances, it has been recently demonstrated that this targeting can modify the clinical outcome of both autoimmune and neoplastic diseases [18,19,20]. Cytotoxic T lymphocyte antigen (CTLA) 4 and programmed cell death receptor (PD) 1 have led to previously detected therapeutic responses in melanomas and non-small cell lung cancer (NSCLC) [21,22,23,24,25,26,27,28]. However, not all patients respond to this therapy, and resistance to this treatment can arise, suggesting that targeting a specific receptor/ligand pair alone is insufficient to defeat the tumor [29,30,31,32]. This finding has triggered the scientific community to look at the use of antibodies against other regulatory receptors, such as killer Ig-like inhibitory receptors (KIRs) and C-lectin type inhibitory receptors (CLIRs) to increase the antitumor efficacy of a given treatment [2,33,34,35,36]. In this context, during the last 30 years a plethora of inhibitory receptors have been discovered, starting from the already mentioned CTLA-4 (identified in 1991 by James Allison) and PD1 (identified in 1992 by Tasuku Honjo) reviewed elsewhere [2,3,7,24,25,28,35,36,37,38,39,40]. Among these, herein we will focus on the leukocyte-associated immunoglobulin (Ig)-like receptor 1 (LAIR1, CD305) [41,42,43]. Overall, this receptor is expressed on almost all leukocytes, and it can regulate different functional activities of healthy T and B lymphocytes, natural killer (NK) cells, and monocytes [41,42,43]. Further, this receptor can regulate the growth and apoptosis of neoplastic cells such as chronic lymphocytic leukemia (CLL) and myeloid acute leukemia (AML) cell lines [44,45]. More recently, the expression of LAIR1 on solid tumors could suggest an additional target for cancer treatment [46,47]

## 2. LAIR1: Identification, Molecular Characteristics and Functional Properties

### 2.1. LAIR1 Discovery and Brief History

An antibody termed 9.1C3 was selected in 1983 for its property of inhibiting the killing of the erythroleukemia cell line K562 exerted by NK cells, mixed-lymphocyte-cultures-induced activated lymphokine killer cells, and monocytes [41,48,49,50]. Afterward, the molecule immunoprecipitated by this monoclonal antibody (mAb) was identified as a glycoprotein associated with the leukocyte common antigen (T200/leukocyte common antigen (LCA)/CD45). Its molecular weight corresponded to 66 and 77kDa proteins. It is of note that anti-LCA mAbs could synergize with the 9.1C3 antibody in blocking natural killer (NK)-cell-mediated cytolysis. More recently, in 1995, four different mAbs: NKTA255, NKTA72, 1F1, and 1B1 were selected based on the inhibition of lysis of the murine mastocytoma cell line P815 mediated by NK cell clones [42]. These mAbs recognized a protein of approximately 40kDa on NK cells under reducing and non-reducing conditions. This molecule has been named p40, and it was apparently different from some other molecules involved in the regulation of NK cell cytolysis, such as the lymphocyte function-associated antigen (LFA)1, CD45 and p58 (later named CD158), the CD94/NKG2 complex, and NKB1/KIR3DL1 [33,34,35]. It appeared that p40 could represent an additional inhibitory receptor molecule of innate immunity in addition to p58, CD94 and NKB1 involved in the NK-cell-mediated recognition of different human histocompatibility antigen (HLA)-I alleles [33,34,35]. Later on, in 1997, LAIR1 was cloned [43]. A DX26 mAb reacted with this receptor of 32KD after de-glycosylation with the N-glycosidase F of a 40kDa glycoprotein [43]. The cloning of LAIR1 allowed to define that all of the above-referenced mAbs could recognize the same cell-surface molecule and its assignment to the cluster of differentiation (CD) antigen 305. CD305 has been described to be differently expressed on leukocyte subsets and involved in the downregulation of several functional activities of immune cells. The main features of the different mAbs recognizing LAIR1 are summarized in Table 1. This table can help to identify the antibody used to characterize the reactivity and functionality, in addition to the potential applications, of reagents recognizing the LAIR1 molecule.

### 2.2. Main Molecular Features of LAIR1

LAIR1 is a member of the immunoglobulin superfamily and in particular of the leukocyte-associated inhibitory receptor family [62]. The region encoding leukocyte-expressed receptors of the immunoglobulin superfamily containing at least 29 genes includes LAIR1 mapping at 19q13.4. Among the other members of the leukocyte immunoglobulin-like receptors (LILRs), LAIR1 bears one extracellular immunoglobulin domain of C2-type and two immune receptor-tyrosine inhibitory motifs (ITIMs). These motifs are responsible for the association of LAIR1 with SH2-containing tyrosine phosphatases 1 and 2 (SHP1 and SHP2), as has been demonstrated by immunoprecipitation with a DX26 mAb and Western blot with anti-SHP1 and anti-SHP2 antibodies [43]. This recruitment leads to the delivery of a negative signal in any LAIR1^+^ cell [41,42,43,48,49,50,51,63,64,65,66]. The molecular features of human and mouse LAIR1 have been previously reviewed in detail by Linde Meyaard [43]. It should be noted that LAIR1 is highly homologous with LAIR2 (about 84%), which lacks the transmembrane and intracellular domain. The LAIR2 molecule (CD306) can be released in the extracellular environment and may compete with the LAIR1 membrane-bound receptor. In addition, mRNA splice variants of LAIR1 have been cloned (LAIR1a, LAIR1b, LAIR1c, LAIR1d, LAIR2a, and LAIR2b); the differences among these isoforms can reside in a few or even a single amino acid in addition to their possible different grades of glycosylation and their specific expression on some leukocyte subsets [43]. In detail, the finding that an anti-LAIR1 rabbit polyclonal antibody identified two different proteins in the myeloid cell line HL60 (45 kD) and Jurkat T cell line (40 kD) would suggest that different degrees of glycosylation as a protein or alternative splicing as an mRNA may exist [54]. Indeed, molecular cloning from Jurkat T cells identified the LAIR1c and LAIR1b isoforms while LAIR1a, LAIR1b, and LAIR1d isoforms have been cloned from HL60. LAIRc differs from LAIR1a by just an amino acid, while LAIR1d lacks the ITIM intracellular domains. Importantly, LAIR1d lacking ITIM may have a dominant negative role in the signaling mediated by ITIM-bearing isoforms of LAIR1. Several other information on molecular features of LAIR1 are reported at [67,68,69]. However, these molecular differences have not been associated with different or specific functional properties yet [43,54]. The study of them could be considered as a major point to discover LAIR1’s biological relevance (see Section 9).

### 2.3. Comparison of LAIR1 Expression and Other Leukocyte Inhibitory Receptors

As mentioned above, LAIR1 is similar to other inhibitory receptors mapping to the same chromosome region and elsewhere [2,10,33,34,35,69,70,71,72,73]. The plethora of inhibitory receptors of leukocytes has represented in recent years a new target for triggering the immune response against cancer [2,10,33,34,35,69,70,71,72,73]. The common feature of all these receptors is represented by the presence of ITIM, the engagement of tyrosine phosphatases, and inhibition of the activation of the activities of the cell bearing the inhibitory receptor [2,10,33,34,35,69,70,71,72,73]. The specificity of function of the different inhibitory receptors is determined by the cellular expression, level of expression, and the ligand recognized [2,10,33,34,35,69,70,71,72,73]. LAIR1 is expressed on both myeloid and lymphoid cells; this can be considered a peculiarity with respect to KIR and CLIR usually clonally expressed on subsets of lymphocytes such as NK and T cells [2,10,33,34,35,69,70,71,72,73]. Apparently, the activating counterpart of both KIR and CLIR can be expressed on the same lymphocyte subsets [69,70,71,72,73]. These molecular isoforms of inhibitory receptors lacking ITIM but bearing an immunoreceptor tyrosine-based activation motif (ITAM) can deliver an activating signal instead of an inhibitory one [69,70,71,72,73]. This activating effect is dependent on the engagement of intracellular tyrosine kinase activities, leading to the activation of effector lymphocytes [2,69,70,71,72,73]. At present, no activating counterparts of LAIR1 have been described. Furthermore, KIR and CLIR interact with some HLA-I alleles, and by definition, the presence of a given isoform of these inhibitory receptors is determined by the expression of a given allele. This implies that cells bearing KIR and/or CLIR are inhibited mainly in an autologous context [69,70,71,72,73]. The first ligands described for LAIR1 were the collagens I and III [74,75,76,77,78,79]. These are essential components of the extracellular matrix, produced mainly by fibroblasts, involved in providing the structural support of the extracellular space of the connective tissues [80,81]. This is a key difference between the two kinds of inhibitory receptors, as LAIR1 can function either in self- or non-self-conditions [2,43]. Further, LAIR1 may reinforce the inhibition mediated by KIR and/or CLIR, or it may counteract the stimulating effects of activating isoforms of KIR and CLIR [70,71,72,73,82]. On the other hand, the known integrin ligands of collagens are the α1β1, α2β1, α10β1, and α11β1 molecules belonging to a subgroup of integrins with an inserted ligand-binding αI domain [81,83]. These collagen ligands are expressed on leukocytes (activated lymphocytes) and mesenchymal and epithelial cells [81,83]. In particular, α1β1 and α2β1 are expressed by long-term-activated lymphocytes and have been previously defined as very-late activation antigens (VLA) 1 and 2 [83]. VLA1 (α1β1 or CD49a/CD29) is typically expressed on memory CD4^+^ and CD8^+^ T cells [83], producing IFNγ in the extra-lymphatic tissues [83]. VLA2 (α2β1 or CD49b/CD29) may show a differential regulation of expression between T helper 1 (Th1) and T helper 2 (Th2) cells, and its engagement with collagen or with agonistic VLA2 mAbs can trigger the CD3/T cell receptor co-stimulation of T cells for IFNγ production [83]. Overall, these integrins binding collagen can display positive effects in the secretion of pro-inflammatory cytokines, while typically the engagement of collagen with LAIR1 leads to inhibitory effects in T lymphocytes [63,66]. Taken together, these findings would suggest that the final outcome of VLA and LAIR1 interaction with collagen could be the result of activating and inhibitory signals in the same effector T cell. The immunoglobulin-like transcripts (ILT) are inhibitory molecules expressed like LAIR1 on monocytes and monocyte-derived cells [84,85,86,87,88]. ILT can recognize a wide range of classical and non-classical major histocompatibility (MHC) class I antigens [85,86,87,88]. In particular, ILT2 (CD85J) is a stronger receptor for HLA-G by two-threefold compared to other MHC molecules [85] and it is expressed on B cells, monocytes, and subsets of T and NK cells [85,86,87,88]. It is of note that ILT2 and ILT4, mainly expressed on monocytes, can inhibit the MHC class I antigen binding of CD8 by competition [85,86,87,88]. This is of particular interest as CD8 is considered a co-receptor of the antigen T cell receptor, and it is consequently involved in the antigen-specific immune response. Overall, ILT2 and ILT4 can inhibit antigen-mediated activation, while this effect is not reported for LAIR1 [85,86,87,88], although the binding of LAIR1 can decrease CD3-mediated triggering with mAbs but not the killing of Epstein-Barr virus-transformed B lymphoblasts by classical cytotoxic T lymphocytes [51,63,66,89]. Similarly to LAIR1, Siglec7, Siglec9, Irp60 (CD300a), and IREM1 (CD300f) are widely expressed not only on lymphoid but also on myeloid cells [90,91]. Siglec7 and Siglec9 can recognize the a2-6-linked sialic acids and a2,8-disialic acid present in ganglioside GD3, the Irp60 interacts with phosphatidylethanolamine (PE) and phosphatidylserine (PS), while the ligand for IREM1 is not discovered yet [90,91,92,93,94]. The wide distribution of these receptors together with the ubiquitous expression of putative ligands would suggest involvement in several tissues and strong regulation mainly of the innate arm of the immune system like LAIR1 [2,90,91,92,93,94,95]. Table 2 summarizes the main molecular features and the cellular expression of the inhibitory receptors in comparison with LAIR1.

### 2.4. Lair1-Mediated Inhibitory Signal and Classical Immune Checkpoint Inhibitors Ctla4 and Pd1

It is generally accepted that CTLA4 interacting with CD80 and CD86 molecules on APCs can effectively downregulate the immune response [95,110,111]. This interaction limits the binding of the co-stimulatory molecule CD28 to the same counter-receptors CD80 and CD86, leading to the reduction/abrogation of the “second signal” to the T lymphocyte. Consequently, the “first signal”, mediated in T cells by the binding of the T cell antigen receptor with the peptide presented in the context of MHC, does not elicit a response in terms of proliferation and effector functions [95,110,111]. This molecular mechanism of competition of CTLA4 and CD28 for the same ligands appears to be different from that occurring between PD1 and its ligands PDL1 and PDL2 [95,110,111]. Indeed, PD1 can directly activate SHP2, which in turn exerts its phosphatase activity on the CD3/TCR-ZAP70 complex and on CD28 [112,113,114]. This dephosphorylation leads to blocking of the first and second signals of activation of T cells. The mode of action of LAIR1 is similar to that of PD1. Originally, it was reported that LAIR1 can be associated with SHP1 and SHP2 in NK cells upon pervanadate stimulation [43,57]. In addition, it has been demonstrated that LAIR-1 can inhibit B cell receptor (BCR)-induced calcium mobilization and tyrosine phosphorylation in DT40 chicken B cells lacking both SHP1 and SHP2 [115]. More interestingly, using a yeast-tri-hybrid system, it has been reported that phosphorylated LAIR1 binds to Csk. This interaction is mediated by the SH2 domain of Csk, docking on the phosphorylated tyrosine residue of the N-terminal ITIM domain of LAIR-1. The phosphorylation of the tyrosine in the N-terminal ITIM of LAIR1 is necessary for linking to Csk [115]. Both the C-terminal ITIM and N-terminal ITIM domains of LAIR1 are necessary to deliver full inhibition of several immune responses and appropriate phosphatase recruitment. However, the mutation of the N-terminal ITIM domain leads to the abolishment of the inhibitory activity [116]. On this basis, the LAIR1-mediated inhibitory signal is different from those from CTLA4 and/or PD1. This point is of interest to hypothesize potential therapeutic approaches in blocking the LAIR1-mediated activity together with those already applied for CTLA4 and/or PD1 (Figure 1).

This could be supported by the finding that the conventional immune checkpoint receptors can be blocked with combinations of antibodies, giving better clinical results compared to the use of single agents [117,118,119,120].

## 3. LAIR1 in Leukocyte Subsets

Hereafter, the experimental evidence of the role of LAIR1 in different leukocyte subsets is reported. Overall, the engagement of LAIR1 delivers an inhibitory signal leading to a decrease in activation through TCR or BCR as well as other activating signals in innate cells, including NK cells or monocytes and monocyte-derived antigen-presenting cells or neutrophils [56,65,75,121,122,123,124,125,126,127,128]. The relevance of LAIR1 in the tumor microenvironment will be considered in a specific chapter, together with the possible ways of regulating the LAIR1-mediated immune escape of tumors.

### 3.1. LAIR1 Expression and Function in T Cells

LAIR1 is expressed on about half of peripheral T cells, and usually at higher levels in CD8 versus CD4 T cells. It is typically expressed at a low level in central memory T cells identified as CD27+ CD45RO+ T cells, while effector cells (CD45RO- CD27-) show higher LAIR1 expression. Its expression is reduced during aging, associated with a reduction in naïve cells as determined in a large cohort of healthy donors (n = 110). The triggering of lymphocytes with a combination of anti-CD3 and anti-CD28 mAbs led to a strong reduction in LAIR1 expression accompanied by an upregulation of lymphocyte activation markers such as IL2Rα, and similarly, antigen-specific T cells were LAIR-null. IFNγ production by peripheral blood mononuclear cells (PBMCs) treated with tetanus toxoid antigen or through the CD3/TCR complex was reduced by the cross-linking of LAIR1 using specific antibodies or collagens [51]. Stimulation through CD3/TCR in the presence of IL2 increased the cell surface expression of LAIR1 in a process that required the activation of p38 MAP kinase and ERK signaling. The apparent discrepancy between the different reports can be related to the different stimuli used to induce the proliferation of T cells [65]. LAIR1 could inhibit T cell proliferation triggered by anti-Vβ antibodies as well as anti-Vδ1 or Vδ2 antibodies [63]. In addition, the TCR signaling was downregulated by decreasing the phosphorylation of the proto-oncogene SRC family tyrosine kinases LCK and LYN, ZAP70, c-Jun N-N terminal kinase 1/2, and p38 [125]. Altogether, these findings indicate that LAIR1 is an inhibitory receptor, whose expression can be up- or downregulated by different combinations of triggers, although the main signal delivered to T cells upon LAIR1 engagement is an inhibitory one.

However, it has been demonstrated in mice that LAIR1 may play a different role in Th1 and Th17 cells [128]. Importantly, the pre-existing Th17 responses to collagen IV (ColIV) were abolished by genetic deletion of LAIR1 in mice. The antagonism of collagen-LAIR1 binding obtained with a chimera molecule composed by the fusion of the collagen-binding portion of LAIR1 and the Fc of IgG (soluble LAIR1/Fc molecule) increased the Th1 response but decreased Th17 responses in mice and in human PBMCs reacting to ColIV. Finally, CD4^+^ Th cells expressing integrin β7 which are prone to transmigrate into tissues express more LAIR1 than β7-negative Th cells [129]. These cells bear at the cell surface other receptors involved in migration, such as CD161/KLRB1/NKRP1A and NTSE (CD73) [130,131]. Overall, these findings suggest that LAIR1 may play roles in the differentiation, activation, and localization of Th T cells. On the other hand, the functional significance in CD8^+^ T cells is prevalently demonstrated in the downregulation of CD3/TCR-mediated triggering and/or alternative molecules such as NKG2D-triggered cytolysis of cellular targets or the production of pro-inflammatory cytokines and intracellular Ca^2+^ mobilization [65,66,82,124]. It has been shown that in patients with chronic hepatitis B (CHB), the T cell number and functionality are increased, although the expression of inhibitory receptors is increased. LAIR1 expression on CD3^+^CD4^+^ and CD3^+^CD8^+^ T cells was markedly decreased, and this effect was evident in hepatitis B e antigen (HBeAg)-positive patients associated with a low level of hepatitis B virus (HBV) DNA. This indicates a negative association of LAIR1 expression levels on T cell subsets with liver inflammation and liver fibrosis parameters [132]. This would suggest that the regulation of LAIR1 expression may be involved in the pathogenesis of CHB, and that LAIR1 can be considered a target for downregulating the immune response.

### 3.2. LAIR1 and NK Cell Activities

As mentioned above, LAIR1 was originally identified in NK cells [41,42,57]. In detail, LAIR1 was involved in the cytolysis of the typical NK cell target K562 exerted by peripheral unstimulated NK cells [41,42,43]. Activation through the CD16/FcγRIIIa of NK cells was inhibited by co-engagement with LAIR1. A similar effect was detected using as a stimulus the triggering through the activating NK cell receptor for HLA-class I antigens. This effect was detectable only after the selection of NK cell clones whose lysis of the P815 target cell line was activated by the antibody directed to the activating form of the KIR or CLIR [63]. Apparently, this activation was not inhibited by anti-CD45 or anti-LFA1 mAb, while CD45 and LFA1 were involved in the binding to target cells [133]. This would suggest that LAIR1 can play a complementary role with other receptors at the cell surface of cytolytic NK cells.

### 3.3. LAIR1 and Antigen-Presenting Cells

Antigen-presenting cells (APCs) including mainly monocytes and monocyte-derived dendritic cells (DCs), are critical players for an appropriate immune response against antigens [134,135]. The most professional APCs are DCs, which can be subdivided in several other subsets [133,134,135]. DC cells from peripheral monocytes lack the typical monocyte marker CD14, while expressing high levels of CD80 and CD86 and neo-expression of CD1a and CD83 [133,134,135]. It has been reported that LAIR1 is present on CD14^+^ and CD14^+^CD34^+^ PBMCs, and these two leukocyte subsets can differentiate into CD14-negative cells upon stimulation with GM-CSF. This monocyte-derived cell population showed good levels of CD80 and CD86 co-stimulatory molecules and lacked LAIR1. The markers analyzed do not clarify whether these cells are full DCs or if they represent an intermediate stage between DCs and monocytes. The engagement on monocytes of LAIR1 with a specific antibody followed by a second reagent in the presence of GM-CSF led to cells with low levels of CD80 and CD86 while still being CD14^+^ and LAIR1^+^. More importantly, the cross-linking of LAIR1 inhibited the GM-CSF receptor-mediated signal transduction in terms of intracellular Ca^2+^ increase [64].

More recently, a detailed analysis of LAIR1 expression on PBMC monocytes and DCs has been reported [136]. LAIR1 was expressed at good levels on all monocytes (higher in the intermediate subset) and plasmocytoid (p)DCs but at lower levels on classical DC1 (CD1c+) and DC2 (CD141+) subsets. It should be noted that LAIR1 was upregulated by pro-inflammatory stimuli such as different TLR ligands, LPS, or IFNα but markedly downregulated by GM-CSF, IFNγ, or IL4 or combinations of these cytokines, leading to the differentiation of monocytes to DCs [137]. Along these lines, it has been reported that the LAIR1 ligand C1q can limit DC differentiation and activation [137]. To our best knowledge, no reports are present in the literature regarding the ability of APCs to present antigens during the engagement of LAIR1. This lack of information should be covered to better plan how to regulate the immune response with an LAIR1 agonist or antagonist. Anyway, it is conceivable that the APCs obtained after cross-linking of LAIR1 should be less efficient in antigen presentation, as they express lower levels of co-stimulatory molecules and lack some markers typical of professional DCs.

### 3.4. LAIR1 Expression on B Cells in Healthy Individuals, Autoimmune Diseases, and Viral Infection

LAIR1 is expressed early during the differentiation of B cells, but it is absent on a fraction of memory B cells and all germinal center B cells, plasmablasts, and plasma cells [122,138]. It is of note that LAIR1 is lost after the stimulation of naïve B cells through the B cell receptor or CD40 ligand. More importantly, the simultaneous cross-linking of LAIR1 and the BCR reduces the intracellular calcium increase triggered by the BCR. This indicates that LAIR1 downregulates BCR signaling, and it can play a key role in the triggering of B cell responses [138]. In this context, it has been reported that LAIR1, CTLA4, and ILT2/CD85j can downregulate the production of IgG and IgE, as well as the percentages of IgG- and IgE-expressing cells [123]. Also, the production of IL8, IL10, and TNFα can be inhibited by the cross-linking of the above-mentioned inhibitory receptors.

Importantly, this inhibitory effect was evident upon the stimulation of B cells with recall antigens such as tetanus toxoid or PPD, CD40L-transfected cells in the presence of IL4, and with LPS plus IL4. These findings indicate that LAIR1, CTLA4, and CD85j/ILT2 can influence IL4-driven isotype switching [123]. The expression and function of LAIR1 was also analyzed in patients with autoimmune diseases such as systemic lupus erythematosus (SLE), mixed connective tissue disease (MCTD), systemic sclerosis (SSc), and rheumatoid arthritis (RA) patients [139]. This inhibitory receptor was expressed at lower levels in SLE and MCTD, but not in SSc and RA compared with healthy donors. Typically, triggering through the BCR, but not via PWM or MALP2, downregulated LAIR1 expression marginally (about 25%). The inhibitory signal delivered by LAIR1 on BCR-mediated intracellular Ca^2+^ increase was less strong in SLE donors bearing fewer LAIR1 molecules at the cell surface. Accordingly, the collagen coated onto culture plates or produced by mesenchymal stromal cells did not inhibit the production of immunoglobulin in SLE patients with a large proportion of CD20^+^LAIR1^−^ B cells [139]. On the contrary, it has been reported that LAIR1 is increased on B cells (both frequency and mean fluorescence intensity) in psoriasis vulgaris (PsV) patients compared to healthy controls [140]. This report did not analyze whether LAIR1 could trigger a stronger inhibitory signal in PsV vs. healthy B cells [140]. Further, more detailed studies in these and other autoimmune diseases should be performed to better define the functional properties of immune subsets expressing LAIR1 to propose this molecule as a therapeutic target. Finally, LAIR1 expression decreased specifically on the memory B cells of HIV1-infected patients prior to antiretroviral therapy (ART): nonlymphopenic, ART-treated nonlymphopenic, or ART-treated lymphopenic. On the other hand, Fas and PD1 expression was increased in some B cell subsets, mainly memory B cells. This observation would suggest that different regulatory pathways of B cells are differently modulated under HIV1 infection [141].

### 3.5. Lair1 in Myeloid and Innate Lymphoid Cells

The analysis of expression of LAIR1 and the homolog-released protein LAIR2 in humans and in mice can give some insights into the relevance and interpretation of the results found in humans and mice. Indeed, some functional activities of a surface molecule can be more easily demonstrated in mice than in humans because the genetic manipulation of mice is a well-established tool to define the biological significance of a specific molecule [142,143,144]. In other words, the generation of genetically determined LAIR1^−/−^ is possible in whole mice, while in humans the overexpression or downregulation of LAIR1 can be achieved in selected cell subsets. With this regard, in humans and mice, the expression of LAIR1 and LAIR2 in monocytes and the differentiation to macrophages have been analyzed using flow cytometry and RT-PCR following stimulation with several stimuli such as LPS, type I and II interferons, and TNFα [58]. LAIR2 can compete with surface LAIR1, as shown by the LAIR2-mediated inhibition of C1q-dependent blocking of monocyte-to-DC differentiation [137]. Importantly, as already mentioned, LAIR2 is only present in humans [63], but LAIR1 and LAIR2 share a similar regulatory genomic neighborhood having opposite functional effects [58]. This may mean that the two molecules are expressed in the same direction. LAIR1 was upregulated during monocyte differentiation to macrophages in both humans and mice. On the contrary, LPS triggered LAIR1 in humans while reducing it in mice. This effect was associated with LAIR1’s dependence on the NF-kB, RELA, and RELB transcription factors in humans, while LAIR1 in mice involved STAT3 and/or STAT5 [58]. These findings would suggest that the molecular mechanisms involved upon stimulation are different in humans and mice, and the lack of LAIR2 expression in mice should be considered when some findings found in mice are transferred to humans.

However, the murine LAIR1 knock-out (KO) model can be useful to better understand the role of LAIR1 because mice do not have a homolog of LAIR2. This avoids the analysis of the presence of LAIR2 in a murine system, and the results obtained with LAIR1 KO should be interpreted strictly due to the absence of LAIR1 [58]. Importantly, these mice infected with *S. aureus* showed dermonecrosis and abscesses that appeared more rapidly and covered areas approximately twice as large as those observed in LAIR1 wild-type (WT) mice. This was accompanied by an increased production of pro-inflammatory cytokines, ROS by neutrophils, and myeloid chemokines together with collagen/ECM pathways. The production of cytokines and chemokines was mainly due to macrophages, which released more than 90% CXCL1 and CXCL2 in LAIR1 KO mice on average compared to WT ones, while CCL2 was released 10-fold more in KO vs. WT. In addition, LAIR1 KO bone marrow-derived macrophages (BMDMs) expressed higher levels both at baseline and after stimulation with the TLR2 agonist PAM3CSK4 of the above-mentioned chemokines compared to LAIR1 WT BMDMs. It should be noted that PAM3CSK4 acts similarly to the *S. aureus* lipotechoic acid, suggesting that this agonist can mimic *S aureus* infection well [145,146]. Finally, it appears that the monocytes and neutrophils of LAIR1 KO mice showed a phagocytosis of *S. aureus* similar to cells from LAIR1 WT mice [97]. Altogether, these findings would support the notion that LAIR1 is a receptor involved in the regulation of the innate immune response against *S. aureus*.

Immunohistochemistry (IHC) assays using the antibody 9.1C3 indicated that the LAIR1 molecule is expressed on about 60% of CD68^+^ liver macrophages, and it appears that this percentage is reduced in biopsies from cirrhotic patients [147]. This is likely due to the higher levels of collagen present during cirrhosis and competition between the anti-LAIR1 antibody and the LAIR1 natural ligand. The subsets of peripheral blood monocytes identified by double immunofluorescence using anti-CD14 and anti-CD16 antibodies indicated that LAIR1 is expressed in healthy donors in classic (CD14^+^CD16^−^) intermediate (CD14^+^CD16^+^) and non-classic (CD14^−^CD16^+^) monocytes. Interestingly, this expression was increased in all of these monocyte subsets in cirrhotic patients [147]. It should be noted that the stimulation of healthy monocytes with LPS or *Candida albicans* did not trigger the upregulation of LAIR1 on monocytes; on the contrary, LAIR1 expression was downregulated as well as in the PMA-differentiated HL60 myelomonocytic cell line [147]. Overall, these results suggest that the increase in LAIR1 expression in the blood of cirrhotic patients may be related to a peculiar combination of still-undefined factors and not to molecular mechanisms involved in some mode of activation of monocytes. Thus, monocyte LAIR1 expression has been proposed as a novel biomarker for the early detection of liver damage caused by cirrhosis.

The silencing of LAIR1 in the THP1 myelomonocytic leukemia cell line reduced partially the expression of LAIR1 [148]. This reduction led to a reduced response to infection with *Helycobacter pylori* in terms of apoptosis and the production of IL8 and IL10. These results suggest that LAIR1 can deliver an anti-apoptotic signal, interfering with the bacterial infection-induced toxicity. The inhibition of production of pro-inflammatory and regulating cytokines would suggest that LAIR1 reduction could modify the behavior of monocyte-like cells. However, the analysis reported was not detailed and performed on healthy monocytes but a leukemia cell line. The expression of LAIR1 at the cell surface as well as the phenotypic alterations present in silenced LAIR1 THP1 cells have not been analyzed. Reasonably, further studies are needed to better define the functional significance of LAIR1 during bacterial infections.

The expression and functional role of LAIR1 in pDCs in humans have been analyzed, suggesting that this receptor can downregulate the production of IFNα triggered by TLR ligands [149]. The level of fluorescence intensity using the 1F1 anti-LAIR1 antibody revealed that pDCs identified as BDCA2^+^ PBMCs exhibited the highest LAIR1 expression among all PBMC subsets. Moreover, IL3 or IFNα or CpG ODN-A (a ligand for ILT9) or combinations of these stimuli led to a marked reduction in LAIR1 expression. Interestingly, this event was accompanied by the upregulation of the NKp44 inhibitory receptor of pDCs. This reduced expression was confirmed in BDCA2+ pDCs in the PBMCs of SLE patients [149]. The stimulation of pDCs with CpG ODN-A led to the production of IFNα and this effect was inhibited by the cross-linking of LAIR1. Similarly, pDCs’ production of IFNα stimulated with SLE anti-DNA immune complexes was significantly inhibited by LAIR1, NKp44, or both inhibitory receptors’ cross-linking. It is of note that the pDCs of *lair1*^−/−^ mice did not produce different amounts of IFNα upon stimulation with CpG [150]. Furthermore, it appears that these mice did not differ in their response to several autoimmune immune disease models such as EAE and colitis [150]. This would suggest differences between LAIR1 function in humans compared to mice. Plasmocytoid DCs are key cells involved in the response to viral infection and they are necessary for the establishment of an appropriate protective immune response [151]. In addition, pDCs can have detrimental effects, leading to the dysregulation of epithelial reparation in autoimmune diseases and skin pathologies, as well as inhibiting megakaryocyte-dependent platelet release [151].

Again, the role of LAIR1 might be important in the regulation of pDC function; the reduction upon IFNα treatment could be considered a molecular mechanism allowing a complete anti-viral immune response or an unwanted exacerbation of a disease like SLE. In this context, it has been reported that the high expression of LAIR1 in myeloid cells can be a biomarker predictive of a deficient type-I-specific IFN (ISG-I) response [152,153,154,155]. Indeed, among the group of 189 cell-surface proteins analyzed, LAIR1 was increased in COVID-19 patients. Autoantibodies against anti-type I IFN have been found in COVID-19 patients and correlated with severe-to-critical COVID-19 [156]. LAIR1 expression is highest in classical monocytes (CD14^+^CD16^−^ PBMCs) at the time of hospitalization, decreases early by 4 days, and predicts an impaired type I IFN response. In COVID-19 patients, anti-LAIR1 autoantibodies have also been detected, again highly specific to severe-to-critical COVID-19 [156]. It is not defined whether LAIR1 plays a causal role or is a consequence of the altered patients’ responses. However, it can be considered a useful biomarker to identify the features of a given patient challenged with SARS-CoV-2 viruses. Eventually, these autoantibodies can influence the immune response together with those directed to several others regulating cell-surface molecules or cytokines and chemokines.

LAIR1 has also been detected in allergic and viral airway inflammation [154,157]. In detail, innate lymphoid type 2 cells (ILC2s) are characterized by the ability to secrete high amounts of IL5, IL9, and IL13 [158] and play a role in triggering airway hyperreactivity (AHR). LAIR1 deficiency exacerbated the pulmonary ILC2-dependent AHR triggered by IL33 and allergen *Alternaria alternata* models in LAIR1^−/−^ mice. Moreover, adoptive transfer experiments confirmed the LAIR1-mediated regulation of ILC2 functionality, and the engagement of LAIR1 by the C1q ligand reduces significantly the AHR in a humanized murine model [157]. Thus, together with other inhibitory receptors such as CD200R, TIGIT, TNFR2, ICOS, ICOS-L, and PD1 [158,159,160,161,162,163], LAIR1 can be considered a good target for therapy to downregulate the AHR in asthma and allergic diseases.

Furthermore, LAIR1 can regulate macrophages’ recruitment to the lungs and their stimulation by polyinosinic/polycytidylic acid to mimic the acute lung injury and acute respiratory distress syndrome (ARDS) in viral pneumonia [154]. The secretion of CXCL10 typical of macrophage overactivation was downregulated by LAIR1. The LAIR1 deficiency present in LAIR1^−/−^ mice augments the lung recruitment of neutrophils, lung resistance, and permeability. Lung macrophages associated with COVID-19 lung inflammation upregulate LAIR1 and immunoregulatory genes including HLA-E, LAPTM5, MAFB, CD68, SIRPA, and HAVCR2 [154]. These findings suggest that inhibitory surface receptors, including LAIR1, can be targets to control the tissue localization and pro-inflammatory functions of myeloid cells in lung injury. The targeting of these receptors could be an additional and/or novel tool to limit injuries due to ARDS in different diseases.

## 4. LAIR2 and Atypical LAIR1 Expression

### 4.1. LAIR2: A Soluble Regulator of Immune Response with Therapeutic Potential, Highlighting the Functional Role of LAIR1

LAIR2 (CD306) can be considered as a soluble form of LAIR1 that functions as a decoy LAIR1 molecule [75,164,165,166,167,168,169,170,171]. LAIR2 in principle binds to collagens with a stronger affinity than LAIR1, leading to a competition with LAIR1-mediated signaling of the cell-surface-expressed inhibitory receptor [75]. This finding has been demonstrated after the synthesis of both recombinant LAIR1 and LAIR2 monomers and dimers, which allowed the determination by surface plasmon resonance of their inhibitory effect in binding to collagen in association with their affinity through surface plasmon resonance [76]. Indeed, the soluble forms of LAIR1 and LAIR2 are monomers, and thus, to understand their function, it is necessary to use the monomer form of the recombinant molecule.

Circulating LAIR2 protein levels are low in healthy individuals but are elevated under inflammation, such as in the serum of patients with autoimmune diseases including RA, ankylosing spondylitis (AS), and autoimmune thyroid disorders, as well as in the urine of pregnant women [76,167,168]. These findings suggest a correlation between LAIR2 expression and inflammation [76]. Soluble LAIR1 (sLAIR1) and LAIR2 can interact with collagen, but LAIR2 abrogates the binding to collagen of LAIR1 better than sLAIR1. This would indicate that LAIR2 produced during inflammatory conditions relieves the inhibitory signal delivered by LAIR1 in effector cells. The final result of this competition between LAIR2 and LAIR1 for collagen depends on the functionality of the immune cells involved (Figure 2). Thus, LAIR2 could be both an enhancer or a suppressor of immune response, impairing the LAIR1–collagen interaction in either effector or regulatory cells. In several solid tumors, elevated LAIR2 levels have been associated with improved prognosis and overall survival, suggesting a role for LAIR2 as a potential therapeutic agent that may reduce LAIR1-mediated immune suppression in the tumor microenvironment [166]. In this context, NC410, a dimeric LAIR2-Fc fusion protein that blocks collagen binding to LAIR1, has shown the ability to stimulate human T cell expansion in a xenogeneic graft-versus-host disease model and enhance T cell-mediated antitumor immunity in a humanized tumor model [166]. In line with this study, recombinant LAIR2-Fc has been shown to block the binding of LAIR1^+^ cells to collagen I/III and the inhibition of release of pro-inflammatory cytokines by stimulated T cells with a combination of anti-CD3, anti-CD28 and CD2 antibodies [172]. Furthermore, the injection of LAIR2-Fc and T cells could exert a strong antitumor effect in subcutaneous xenograft murine models of the A431 or Bx-PC3 cell lines. More relevantly, LAIR2-Fc potentiated the antitumor effects of the PD-1/PD-L1 immune checkpoint blockade [172]. Finally, it has been shown that the LAIR2-Fc protein can affect the binding of murine cells to murine collagen and the growth of tumors in immunocompetent mice. This was accompanied by an increase in IFNγ and granzyme B production by T murine cells [172]. Collectively, these findings support the development of LAIR2 as a novel immunotherapeutic target. Accordingly, a first-in-human Phase I clinical trial (NCT04408599) is currently underway in ovarian, gastric, and colorectal carcinomas to assess the safety and tolerability of NC410.

### 4.2. Expression of LAIR1 on Stromal Cells and Soluble Lair1 Presence in Some Diseases

Further complicating this intricate scenario is the reported presence of LAIR1 on human fibroblast-like synoviocytes (FLSs) in RA [173]. The expression of LAIR1 was shown using the antibody 91.C3 in both in vivo specimens and in cultured FLSs positive for the intracellular fibroblast marker vimentin. Some of these FLSs were also faintly positive for CD45, possibly supporting the notion that the FLSs bear the common leukocyte antigen or that some leukocyte-like cells were present. However, a detailed characterization of FLSs has not been reported to further clarify the presence of leukocyte vs. mesenchymal markers. The use of the MH7A, a transformed human synovial cell line, overexpressing the LAIR1 receptor (LV-LAIR1) by lentiviral transduction allowed the demonstration that TNFα can specifically reduce LV-LAIR1^+^ MH7A cells’ invasiveness and led to the release of LAIR1 from the cell surface. This last event was confirmed in primary LAIR1-transfected FLS [173]. Importantly, this TNF-dependent shedding of LAIR1 was inhibited by serine protease inhibitors. The production of IL6 and IL8 triggered by FLSs treated with TNFα was markedly reduced in LV-LAIR1^+^ MH7A cells. Along these lines, mRNA expression of MMP13 was also reduced upon TNFα treatment in LV-LAIR1+ MH7A FLSs compared to control MH7A FLSs. This might imply that pro-inflammatory cytokines exert a low effect when acting on FLSs expressing LAIR1. The role of sLAIR1 has not been analyzed in detail. In this context, the reduction in surface LAIR1 on FLSs mediated by TNFα should evoke a less inhibitory effect on FLSs favoring the inflammation process. Overall, a potential role of LAIR2 in the synovial microenvironment has not been considered yet.

The analysis by ELISA of supernatants of PMA, PHA, or anti-CD3-mAb-triggered lymphocytes indicated the presence of sLAIR1 [174]. The serum levels in healthy individuals were quite low compared to those with hemorrhagic fever with renal syndrome and after kidney transplant. sLAIR1 was further increased in patients with transplant rejection compared to those without rejection. Altogether, these findings indicate that sLAIR1 can be considered a biomarker during inflammation, although its functional role is not clear [174]. LAIR2 is usually produced by CD4^+^ T cells better than CD8+ T cells [167]. Further, among CD4+ T cell subsets, mainly naïve CD4^+^ T cells after stimulation can release LAIR2. On the other hand, the production of sLAIR1 is lower than that of LAIR2 and in several instances is undetectable. The cellular source of sLAIR1, in addition to LAIR2, found in urine and plasma, has not been determined, but again, sLAIR1 is increased in RA patients but not in osteoarthritis. Apparently, the fold of the increment in sLAIR1 was greater than that of LAIR2 in plasma, while both proteins were increased in synovial fluid and urine of RA. The functional role in RA patients of these two soluble inhibitory receptors has not been investigated, although it is conceivable that whatever the source of these receptors, they can compete with the surface LAIR1 which in turn can be reduced by the release of the soluble molecules. Overall, the final results will be a reduction in the LAIR1-mediated inhibitory signal, and conceivably this will result in increasing inflammation [167].

Recently, soluble LAIR1 was compared to the intensity levels of expression on different immune cells in pediatric patients suffering from the autoimmune vasculitis Kawasaki Disease (KD) [175]. It has been found that sLAIR1 is increased in patients with KD compared to healthy controls. More interestingly, the sLAIR1 was further increased in KD patients with coronary artery aneurysms. This increase has been interpreted as a compensatory response to the vascular inflammation and increased immune response. The increase in sLAIR1 might be related to the decrease in LAIR1 at the cell surface and consequent reduction in immune regulation. Of course, this scenario does not consider the expression of other inhibitory receptors on immune cells, as well as the possible imbalance among the different lymphocyte and neutrophil subsets present in KD patients. In other words, an increase in a soluble molecule could affect the overall response in a specific disease such as KD in the context of the inhibitory and activating receptors present on immune cells and their specific function. Furthermore, in this report, no analysis of LAIR2 has been reported, nor its phenotypic and functional features of immune cells [175]. This implies that further analyses may reveal the relevance of the sLAIR1 molecule in KD.

## 5. Expression of LAIR1 on Tumor Cells

The presence of any inhibitory receptor on tumor cells could be used as a therapeutic target for three main reasons: (1) if expressed at good levels, they can be a target for therapy with antibodies conjugated with drugs or radiolabeled isotopes (as in antibody–drug conjugates such as anti-CD30 vedotin or anti-CD33 ^225^Ac-Lintuzumab) [176,177,178]; (2) the antibody could favor recognition by immune cells of tumor targets impairing binding with the ligand (as happens with classical immune checkpoint inhibitor (ICI) antibodies directed against CTLA4 or PD1) [26,179]; (3) the antibody against the inhibitory receptor can deliver a negative signal into the tumor target cell, leading to the inhibition of cell proliferation or the induction of cell death (Figure 3).

Hereafter, we will report and discuss the experimental evidence supporting the possible use of LAIR1 as a target for tumor therapy in both hematological and solid neoplasia.

### 5.1. LAIR1 and Hematological Malignancies

#### 5.1.1. LAIR1 on Leukemic Myeloid Cells

LAIR1 is expressed on the neoplastic counterpart of healthy leukocytes such as myeloid and lymphoid cells [42,57,63,64,76,82,127]. Both established tumor cell lines and primary tumor cells express this inhibitory receptor. It has been reported that LAIR1 can impair the proliferation of myelomonocytic cell lines, leading to cell apoptosis interfering with NF-kappa B nuclear translocation [45]. This apoptotic effect was not affected by the caspase-1 and caspase-8 specific inhibitors, at variance with FAS-mediated programmed cell death. Furthermore, the engagement of LAIR1 on primary acute myeloid leukemia (AML) can inhibit the proliferation induced by granulocyte monocyte-colony stimulating factor (GM-CSF) [45]. This proliferation is conceivable due to the GM-CSF-mediated phosphorylation and activation of Akt1/protein kinase B alpha and intracellular Ca^2+^ flux that are inhibited by LAIR1 engagement [45]. These findings have been detected using a specific mAb and an anti-mouse antibody, leading to the cross-linking of LAIR1 molecules at the cell surface of primary AML [45]. Importantly, without cross-linking of the primary antibody, no effect was detected. Collagens are ligands for LAIR1 and there are multiple sites on collagen for ligation with LAIR1 [74,78]. These multiple binding sites could engage several LAIR molecules, mimicking the cross-linking of LAIR1 mediated by an anti-mouse antibody [74]. Along these lines, it has been reported that LAIR1 agonists have an anti-LAIR1 antibody termed NC525 [60]. The antibody NC525 triggers the cell death of AML blasts and leukemic stem cells (LSCs). This apoptosis is linked to the block through SHP-1 activation of aberrant PI3K/mTOR activity. Consequently, the constitutive activation of the MAPK present in AML blasts is impaired, and the proliferation signal promoted by Akt and NF-kB is decreased. This leads to the deactivation of BCL-XL, triggering an apoptotic signal through caspase-7 and PARP and eventually programmed cell death. Also, NC525 triggered the dephosphorylation of ERK1/2, GSK-3β, and JNK. The inhibitory signals mediated by LAIR1 were regulated by the level of expression of this inhibitory receptor. Indeed, high levels of expression have been detected on LSCs and blast cells but not on healthy hematopoietic stem cells (HPCs). This renders LAIR1 a putative anti-leukemic target. Importantly, in in vivo murine models including cell- and patient-derived xenografts, the NC525 anti-leukemia effect was increased in the presence of collagen. Moreover, NC525 significantly improved the activity of azacitidine and venetoclax, the typical standard therapy for AML [60]. It is of note that the effect mediated by LAIR1 is complementary to those of venetoclax, which blocks the anti-apoptotic B cell lymphoma 2 (Bcl2) protein, and azacytidine as an inhibitor of DNA methyltransferase.

Previously, it has been reported that the LAIR1 receptor is essential for the growth of human leukemia cells [180]. This has been stated on the basis of individually silencing a plethora of ITIM receptors in AML cell lines. It has been shown that the strong reduction in LAIR1 expression led to an impairment of cell proliferation in a series of AML cell lines, including MV4-11, THP1, and U937. Similarly, the lentivirus-mediated silencing of other inhibitory receptors such as KIR3DL1, PECAM1, LILRB1, LILRB3, and LILRB4 resulted in a marked inhibition of cell proliferation, further supporting the notion that several inhibitory receptors can be essential for AML proliferation. On the other hand, LILRB5 and Siglec 7 silencing did not exert a clear inhibitory effect. Altogether, these findings would suggest that different inhibitory receptors play a positive or negative regulatory role in AML cell growth. This information should be considered when therapy is planned targeting these inhibitory receptors. In addition, the in vivo bone marrow engraftment of MV4-11 leukemia cells in transplanted NOD/SCID-IL2RG (NSG) mice was partly impaired using silenced LAIR1 cells, further supporting the need for LAIR1 in optimal leukemia cell growth. This concept was confirmed in LAIR1-null mice using mouse AML models such as MLL-AF9 and AML1-ETO9 [180]. Importantly, the growth of LAIR1-null leukemic cells was almost abolished after the second transplantation, and the frequency of functional AML stem cells in a LAIR1-null primary MLL-AF9 model mice was 1/53 that found in control AML mice.

This would suggest that LAIR1 supports the stemness of AML cells. Furthermore, it has been shown that SHP-1 is a negative regulator in normal myeloid progenitors, but it is a positive mediator in AML stem cells, preventing their exhaustion [180]. Importantly, this positive effect on AML stem cells is linked to the Grb2-binding-independent SHIP1. Indeed, LAIR-1 through Ca^2+^/calmodulin-dependent protein kinase I (CAMK1) and cAMP response element-binding protein (CREB) was a key receptor to sustain LSCs. On the other hand, this receptor was dispensable in healthy LSCs. Overall, these reports support the notion that LAIR1 is a key molecule for AML LSCs and AML blast cells. The significance of other receptors able to deliver a negative signal to AML cells is still to be elucidated, and future research on the use of combinations of agonists of these receptors could give new insight to promote novel anti-AML therapies.

#### 5.1.2. LAIR1 on Leukemic/Lymphoma B Cells

Some reports have demonstrated the expression of LAIR1 in acute lymphoblastic leukemia (ALL) [181,182,183]. The analysis of a cohort of 6 T-ALL and 36 B-ALL revealed an apparently reduced level of expression of LAIR1 in these patients. In addition, the LAIR1 expression level was not statistically associated with the ALL risk factors, previous treatment, or minimal residual disease (MRD). A possible role of LAIR1 in B-ALL cells was supported by the finding that a high expression of LAIR1 together with CD300A and PECAM1 has been found in ALL cells but not in healthy pre-B cells. Importantly, the systematic analysis of 109 ITIM-bearing receptors in humans (63 typically expressed in B cells) indicated that ALL characterized by the Philadelphia chromosome (Ph^+^) showed a stronger expression of LAIR1, CD300A, and PECAM1, among others. Dividing the patients of two clinical trials (P9906, ECOG 2993) on the basis of the median expression of these three different receptors, it has been revealed that high median expression predicts overall survival and relapse-free survival. Experiments with pre-B cells isolated from the bone marrow of *Pecam1*^−/−^, *Cd300a*^−/−^, and *Lair1*^fl/fl^ mice and wild type controls and expanded with IL7 or transformed with *BCR-ABL1* to model human *Ph*^+^ ALL showed the functional relevance of the above-mentioned inhibitory receptors. The loss of ITIM receptors had no significant effects on the cell proliferation or survival of normal pre-B cells. On the contrary, pre-B ALL cells lacking ITIM-bearing receptors underwent cellular senescence and a block of the cell cycle and did not form colonies while expressing cell cycle checkpoint molecules, as well as an increase in reactive oxygen species (ROS). The ablation of LAIR1 surface expression resulted in the hyperactivation of Syk, SRC kinase, and Erk leading to cell death in vitro, remission of leukemia growth in vivo, and prolonged survival in transplant recipient mice. This hyperactivation is typically present as a condition to trigger the negative selection of autoreactive pre-B cells [184] during B cell development. These findings support the idea that LAIR1 can be a regulatory molecule that is important, likewise other inhibitory receptors, to allow the correct differentiation of B cells. The deletion of SHIP1 and SHP1 led to increased p53 cell cycle checkpoint molecules’ expression and cell cycle arrest in G_0/1_ phase, with a strong reduction in colony formation. This also suggested that the targeting of some phosphatases, and not only of ITAM-activating molecules, can be a therapeutic tool for B-ALL leukemia. Importantly, this event does not happen in chronic myeloid leukemia (CML) because the deletion of LAIR1 and different phosphatases does not lead to negative selection as for pre-B cells [184].

Similarly, LAIR1 is expressed on B cells of chronic lymphocytic leukemia (CLL) [44,185,186,187,188]. This expression is lower compared to peripheral blood B cells of healthy donors, and in some CLL patients this expression is negligible. Indeed, LAIR1 is mostly absent in high-risk (HR) CLL, and the intensity of expression is associated with the stage of disease. The activation of p38 and JNK triggered by B cell receptor engagement is impaired by LAIR1. Furthermore, LAIR1 markedly inhibits the constitutive and BCR-induced Akt activation as well as the nuclear translocation of NF-kB activation, preventing the proliferation of B CLL cells [44]. The effect of LAIR1 was detectable in HL CLL expressing low levels of LAIR1, but not in LAIR1-negative samples [44]. Altogether, these findings support the notion that, in some HR CLL, the absence of LAIR1 leads to the lack of a point of control of cell proliferation. On this basis, it can be suggested that the binding of LAIR1 to collagens can evoke a negative signal for CLL proliferation. This signal could be absent in HR CLL, favoring their growth. It is to be determined which other B CLL surface molecules can concur with or counteract the LAIR1 effect. Accordingly, it has been reported in a consecutive cohort of 311 CLL patients that LAIR1 was inversely related to CD38 and the expression of LAIR1 was significantly lower in patients with Binet stage B or C disease than in healthy controls. Also, this decrease in expression was associated with HR cytogenetic abnormalities or unmutated immunoglobulin heavy chain variable region genes. In addition, LAIR1 showed an independent role in predicting the time to first treatment (TTFT), although LAIR1’s ability to predict the survival of CLL patients is still to be determined [188]. An inverse relationship regarding the expression between LAIR1 and the CD200 receptor in CLL has recently been reported [185]. Indeed, LAIR1 was more highly expressed than CD200 in low-risk CLL patients and vice versa in HR patients. In this study, the authors claimed that Treg cells were increased in CLL patients and that this was positively associated with high CD200 expression, while there was a negative association with LAIR1 [185]. It has been reported that CD200-CD200 receptor interaction can lead to either pro- or antitumor effects. Further, the protumor effect of CD200 can be associated with the positive stimulation of Treg as reported by using CD200 blocking antibodies [189,190]. To our knowledge, it is still to be shown whether LAIR1 can trigger the activity and/or proliferation of Tregs.

High LAIR1 expression in diffuse large B cell lymphoma is associated with a poor outcome [191]. Also, BCL2, CD39, or CD103 are related to this low overall survival as a function of time. Importantly, it is possible to formulate a risk score for DLBCL patients considering BCL2, BCL6, CD11c, and LAIR1 expression in DLBCL. LAIR1 is a good marker for the assessment of minimal bone marrow involvement by B-non-Hodgkin lymphoma (B-NHL) [192].

### 5.2. LAIR1 Expression and Function on Solid Tumor Cells

Typically, LAIR1 is a surface molecule expressed by leukocytes, but there are some reports claiming it is expressed by cancer cells [193,194,195,196,197,198,199,200,201].

It has been described that anti-LAIR1 antibodies can react with epithelial ovarian carcinoma (EOC) specimens, but not with ovarian tissue near the tumor. Also, some EOC cell lines, including COC1 and HO8910, express higher mRNA for LAIR1 than OVCA433, OVCAR-3, HEY, CAOV3, A2780, and SKOV3 [193]. However, the flow cytometry analysis indicated that COC1, but not HO8910, reacted with the anti-LAIR1 mAb DX26. It should be noted that the silencing of LAIR1 in HO8910 led to an increase in proliferation, colony formation, and matrix invasion. LAIR1 expression correlated with the tumor grade [193]. LAIR1a and LAIR1b were both expressed in HO8910, and LAIR1 suppressed cell growth by inhibiting the PI3K-AKT-mTOR axis [52]. The transfection of LAIR1 into SKOV3 led to an inhibition of proliferation, migratory abilities, and the triggering of apoptosis [52]. On the other hand, the expression of LAIR1 has been detected in breast carcinoma (BC) and some BC cell lines such as SKBR3 (HER2^+^) and MDA-MB-231 (HER2-) but at very low levels in MCF7 and MCF10A [194]. Surprisingly, the silencing of LAIR1 in both of these cell lines reduced LAIR1 expression by more than 60%, and this reduction was accompanied by less cell proliferation and a lower invasion ability. Also, high expression of LAIR-1 in BC was associated with shorter patient survival, as determined by a detailed bioinformatic analysis [194]. These findings are in line with what was reported in renal cell carcinoma (RCC) [56].

Indeed, it has been found by PCR that LAIR1 mRNA was upregulated in human RCC tumor tissues compared to the next non-tumor renal tissues. The presence of soluble LAIR1 was evidenced in tissue-exudative extracellular vesicles (Te-EVs) in RCC regions compared to the next non-tumor regions. The overexpression of LAIR1 in the ACHN cell line determined an increase in Akt activation and proliferation, while the silencing of LAIR1 in Caki-2 caused the opposite effect. These effects on tumor cell growth were confirmed in mouse xenografts, and more importantly, RCC patients with higher LAIR1 expression showed a reduced OS and PFS [56]. These findings would suggest that LAIR1 can provide a positive signal in RCC. However, these findings were not supported by the direct evidence that the engagement of LAIR1 in RCC can trigger cell activation. As mentioned above, the reported observations with LAIR1^+^ myeloid leukemia are in contrast with the findings reported in RCC [198]. However, the experimental setting is completely different: in myeloid cells, the cross-linking of LAIR1 is triggered by a second reagent recognizing an anti-LAIR1-specific monoclonal antibody, while for RCC this setting has not been assessed. Further, the authors claimed that they did not detect reactivity at the cell surface by flow cytometry with the rabbit polyclonal anti-LAIR1, while myeloid cells reacted well with an anti-LAIR mAb. Overall, the direct effects of LAIR1 cross-linking on solid tumor cells should be tested to better support the notion that LAIR1 triggers a positive signal in some tumor cell lines.

LAIR1 has been detected in cervical carcinomas (CCs) in IHC assays [199] and its expression was associated with tumor size, pathological differentiation, T histological classification, clinical stage, and the involvement of regional lymph nodes. Further, the stable transfection of LAIR1 determined a reduction in proliferation and apoptosis. It has not been investigated how the neo-expression of this inhibitory surface molecule can deliver a regulating signal, and the phenotypic features of CC LAIR1^+^ expressing cells have not been analyzed in detail [199].

Also, LAIR1 has been found on osteosarcoma tumor specimens by IHC and at a variable level of expression by WB in osteosarcoma cell lines such as MG63, SAOS2, U2OS, HOS, and SJSA-1 and healthy osteoblast hFOB1.19 cells [53]. The overexpression of LAIR1 in the HOS cell line led to a marked inhibition of cell migration and epithelial–mesenchymal transition (EMT) impairing the glucose transporter Glut1 expression [53]. Although it is unclear by which molecular mechanism the upregulation of LAIR1 can influence the key processes of HOS cells, the resulting effect should be an inhibition of several protumorigenic events. Again, the reactivity by flow cytometry and the direct analysis of LAIR1-mediated signaling have not been assessed in osteosarcoma cell lines.

The expression of LAIR1 in low-grade glioma (LGG) and glioblastoma (GBM) tissues was notably higher compared to its healthy counterpart [200]. The bioinformatic analysis of single-cell RNA sequencing data showed that LAIR1 was better expressed in glioma cells and macrophages/monocytes but less in astrocytes, endothelial cells, neutrophils, and T cells present in human brain cancer tissues [200]. Furthermore, the analysis of LAIR1 expression in human U87, U251, T98G, and U138 cells, mouse GL261 cells, and rat C6 revealed that LAIR1 was found to be present in U251 and T98G glioma cell lines, but not in U87 or C6 glioma cells. Using an orthotopic murine model with the transfected murine cell line GL261, it has been shown that LAIR1 overexpression is associated with larger tumors than control non-transfected GL261 cells. This effect was associated with a stronger component of immunosuppressive macrophages, suggesting that LAIR1 overexpression can skew the polarization of myeloid infiltrating cells [200].

In hepatocarcinoma cells (HCCs), it has been shown that LAIR1 increases PD-L1 expression through the GSK-3β/β-catenin/MYC/PD-L1 pathway, promoting the immune evasion of tumor cells. Importantly, the targeting of LAIR1 helped to increase the killing effect of CD8^+^ T cells [196]. Accordingly, LAIR1 expression was associated with poor HCC differentiation and a worse OS, suggesting that LAIR1 may be an independent prognostic parameter in HCCs [201]. Table 3 summarizes the findings and functions of LAIR1 reported in solid tumors.

## 6. LAIR1 Expression and Function in the Tumor Microenvironment

LAIR1 can play a regulatory role in several types of immune responses, possibly shaping the inflammatory response to several stimuli and maintaining tissue homeostasis by interacting with its widely expressed ligands. Study of the role of an inhibitory molecule in the context of a neoplastic disease is important as its targeting can be used to modify the antitumor immune response and the tumor microenvironment (TME) as has been shown for the therapeutic actions of ICIs [15,26,100,117]. The relevance of LAIR1 in the TME is related to the finding that mesenchymal stromal cells producing collagen can be considered a key cellular player involved in the regulation of the antitumor immune response [202,203,204]. The fact that some collagens are ligands for LAIR1 implies that interfering with LAIR1–collagen binding in the TME can downregulate the LAIR1-mediated inhibition of antitumor immune cells [69].

Bioinformatic analysis of TCGA datasets indicated that the high tumor mRNA expression of MMPs, collagen I, and LAIR1 has a worse OS, suggesting a possible relationship between MMP activity and LAIR1-mediated inhibition of the antitumor T cell response [205]. In line with this hypothesis, the collagen fragments generated by MMP1 and MMP9 can inhibit CD3 signaling and IFNγ production, and the use of a LAIR2-Fc fusion protein can restore the CD3-mediated activation [205]. This suggests that MMP activity can regulate LAIR1-mediated signaling and the potential use of LAIR2 to trigger T cell activities [205]. The remodeling of the TME targeting the collagen/LAIR1 interaction together with TGFβ and PDL1 revealed that the enhanced tumor infiltration and activation of CD8^+^ T cells could be detected in murine models of colon and mammary carcinoma [206]. This effect was accompanied by the repolarization of suppressive macrophage populations, leading to long-lasting tumor protection. These results indicate the advantage of targeting both the matrix components and ICI to increase the efficacy of immunotherapy [206].

Along these lines, it has been shown that LAIR1 is associated with stroma and strongly expressed in several human tumors, mainly on myeloid CD68^+^ SMA^+^ cells with suppressive functional features in human tumor and tumor murine models [59]. The use of a humanized anti-LAIR1 antibody, NGM438, can interfere with myeloid and T cell interactions with collagen, eliciting myeloid inflammation and allogeneic T cell responses in humans. Also, a mouse reactive NGM438 antibody sensitized a refractory KP 344SQ model in mouse lung tumors to anti-PD1 therapy, triggering CD8^+^ cell infiltration and inflammatory responses [59]. This model has been chosen because the KP 344sQ cells can downregulate immune responses related to LAIR1-mediated inhibition [165]. These findings clearly indicate that the blocking of collagen-mediated inhibitory effects can relieve the lack of response to ICI and that LAIR1 can play a key role in the generation of the immunosuppressive TME found in the stromal regions of several tumors [203,207,208]. Along these lines, the reported T cell exhaustion through SHP1 triggered by collagen via LAIR1 [165] and the reduction in collagen deposition through LOXL2 enzyme suppression favors T cell tumor localization, abrogating the resistance to anti-PDL1. Indeed, the LAIR2 overexpression competing with the LAIR1 signal can sensitize murine models of lung tumors to anti-PD1 treatment [165].

LAIR1 expression in the TME has been shown on M2 macrophages, neutrophils, and T regulatory cells in association with the presence of the lymphocyte-specific protein (LSP) 1 in a series of gliomas [209]. This expression is associated in macrophages with the expression of PD1. This finding suggests that multiple inhibitory receptors can be involved in shaping the TME [209]. In addition, it has been reported that the regulatory T cells in leukoplakia, a pre-cancerous alteration, and head and neck squamous cell carcinoma tissues express LAIR2, favoring an immunosuppressive environment for tumor growth [210].

It should be noted that the analysis of gene signatures from human samples of mesothelioma identified monocyte-derived tumor-associated macrophages with a high expression of some genes, including LAIR1, in addition to TREM2, STAB1, MARCO, and GPNMB [211]. Altogether, these findings indicate that in different experimental systems the role of LAIR1 in shaping the TME and this receptor can be considered a suitable target for immunotherapy, evaluating its expression together with the presence of other regulatory receptors [212,213,214,215].

The expression of LAIR1 on tumor cells should be considered together with the expression of other inhibitory receptors of immune cell functions [212,213,214]. Indeed, it has been shown that CTLA4 can be expressed by human melanoma cell lines and patients suffering from melanoma [214,215,216]. Immunotherapy with a specific antagonistic/blocking antibody against the same inhibitory receptor expressed on both tumor cells and antitumor lymphocytes could reduce the inhibitory signal favoring tumor cell growth while triggering the antitumor effect. The net effect should be a positive one, but this is related to the specific function of the LAIR1-expressing leukocyte subset. For these reasons, the use of antagonistic/blocking antibodies should be analyzed functionally in great detail prior to their use for therapy (Figure 4).

## 7. LAIR1: A Double-Edged Sword in *Plasmodium falciparum* Immune Evasion and Host Defense

To complete the scenario and further reinforce the idea that LAIR1, as well as other inhibitory molecules, can play a key role in immune reactions, the ability of LAIR1 binding has been reported with some proteins expressed by infected erythrocytes with malarial parasites regulating parasite immune escape [217,218]. Malaria is an infectious disease caused by protozoan parasites of the *Plasmodium genus*, with *P. falciparum* being the most virulent species, responsible for the majority of malaria-related deaths [218]. Infected erythrocytes express surface proteins encoded by a highly polymorphic multigene family, including members known as RIFINs [203,204]. Some RIFINs can bind inhibitory receptors, such as Leukocyte Immunoglobulin-like Receptor B1 (LILRB1) and LAIR1 [219]. Specifically, binding between RIFINs and LILRB1 has been shown to inhibit the activation of LILRB1-expressing B cells and NK cells [219]. In contrast, the interaction between RIFINs and LAIR1 has not been conclusively shown to inhibit immune cell activation. Interestingly, in a cohort of malaria-exposed African individuals, antibodies containing LAIR1 inserts with somatic mutations have been identified. These mutated LAIR1-containing antibodies bind strongly to RIFIN on infected erythrocytes, potentially blocking the interaction between RIFIN and the native LAIR1 receptor. Wild-type LAIR1, on the other hand, binds only weakly or not at all to these RIFINs [219,220,221,222]. Taken together, these findings highlight the dual role of LAIR1 in malaria infection: first, as a target of the parasite’s immune evasion strategy, where *P. falciparum* uses RIFINs to bind LAIR1 on immune cells, inhibiting their activation; and second, as a mimetic target for host-derived mutated antibodies, which incorporate segments of LAIR1 to block this interaction and counteract the immune evasion mechanism. These observations raise intriguing questions about whether LAIR1-mimicking antibodies, which are naturally produced by individuals exposed to malaria, could inspire novel vaccine strategies. By inducing similar antibodies that block RIFIN-LAIR1 interactions, it may be possible to enhance the immune recognition of *Plasmodium falciparum*. However, given LAIR1′s crucial role in maintaining immune homeostasis, targeting this pathway must be approached with caution to avoid potential immunopathology or autoimmunity.

## 8. LAIR1 Expression on Hematopoietic Cell Precursors and Role in the Regulation of Hematopoiesis and Cell Differentiation

We have already analyzed the role of LAIR1 on discrete leukocyte subsets of either myeloid or lymphoid lineage in the corresponding Section 3.1, Section 3.2, Section 3.3, Section 3.4 and Section 3.5. It is evident that LAIR1 is expressed on almost all leukocytes after isolation while its expression can be downregulated upon differentiation. This suggests a role of this molecule mainly on naïve and undifferentiated cells. In this context, the main point to be considered is that SHP1 associates with LAIR1 [54] and that SHP1 is a major phosphatase involved in the regulation of hematopoiesis as shown in motheaten and viable motheaten mice [223,224]. Indeed, these mice develop a severe autoimmune and immunodeficiency syndrome with a high proliferation rate of all hematopoietic cells related to the altered SHP1 activity. Antibodies to LAIR1 can activate SHP1 and this effect may indicate that LAIR1 modulates hematopoiesis. LAIR1 is well expressed on CD34+ hematopoietic progenitor cells [225] and in particular on immature megakaryocytes (MKs) [226]. Importantly, the cross-linking of LAIR1 on immature MKs can inhibit the generation of MKs induced by a cocktail of cytokines containing thrombopoietin [226]. These findings suggest a key role of LAIR1 for hematopoietic cell precursors. As indicated in Section 5.1.1, the expression of LAIR1 appeared to be stronger on leukemic than healthy stem cells [60]. This finding may indicate that targeting of LAIR1 with agonistic antibodies can spare healthy HSCs as reported [60]. Altogether, these reports claimed the key presence of LAIR1 on HSCs and its association with regulating phosphatases involved in HSC differentiation and the regulation or proliferation of stem cells. The relevance of LAIR1 on HSC self-renewal and lineage differentiation is not defined. It might be hypothesized that growth factors involved in the proliferation and differentiation of HSCs into specific lineages can influence the level of expression of LAIR1 differently. The study of this possibility should be faced before using the targeting of LAIR1 in hematological malignancies.

## 9. Future Research to Identify the Knowledge Gaps of LAIR1 Function and Its Therapeutic Targeting

Several knowledge gaps on LAIR1 functions remain to be filled to plan the use of some therapeutic tools such as agonist/antagonist antibodies or Fc chimeric molecules. Indeed, the wide expression of LAIR1 on immune cells with different functionalities is the first main problem to consider. The presence of several inhibitory isoforms of LAIR1, including LAIR1a, b, and c, together with the possible expression of the non-inhibitory LAIR1d isoform lacking ITIM is another key point to be addressed (see Table 4).

A systematic analysis of the expression of each isoform on healthy and pathologic cells can shed some light on the biological significance of LAIR1 in physiological functions and in several inflammatory and neoplastic diseases. The generation of appropriate tools, such as antibodies reacting with specific LAIR1 isoforms, could shed some light on their function. The type and level of intensity of signal delivered through LAIR1 by binding with its different ligands has not been investigated thoroughly. This could play a role in the regulation of inflammatory processes and strong immune suppression. Further, LAIR1-mediated effects upon interaction with collagen should be considered together with the presence of LAIR2 and microvesicle- and/or exosome-associated LAIR1 isoforms. This means that the result of LAIR1 engagement due to a therapeutic tool can be different in the presence of non-surface-associated LAIR molecules. Focusing on the potential targeting of LAIR1 in cancer and therapeutic applications, two aspects should be considered. First, the effect of LAIR1 agonism in hematological tumors should spare the targeting of antitumor immune cells. Second, LAIR1’s role in solid tumors is not well-defined. The first point implies the need to study in detail the physiology of LAIR1 on healthy hematopoietic stem cells to target mainly/only neoplastic stem cells. Indeed, the degree of expression of LAIR1 on healthy and neoplastic HSCs is clearly overlapping and it is not clear whether the different anti-LAIR1 antibodies recognize overlapping epitopes or not [60,225]. Furthermore, the presence of the other isoforms of LAIR1 or LAIR2 can regulate the differentiation of HSCs. This suggests an analytical analysis of reactivity of anti-LAIR1 mAb and their functional effects on HSCs. Only these analyses might predict the precise effect of agonist antibodies delivering negative signals on the proliferation of target cells. Also, the role of the LAIR2-Fc chimera should be analyzed in this context. The use of LAIR2-Fc chimeras to treat some carcinomas with the intent of blocking the inhibitory effect in antitumor immune cells can influence the correct development of HSCs and eventually the specific antitumor response.

The second point is much trickier to solve; indeed, the targeting of LAIR1 with agonists or antagonists should be considered applicable in solid tumors but different tools should be selected depending on the solid tumor. The study of the role in solid tumors of LAIR1 is only at its beginning. It should be clarified whether the high LAIR1 expression associated with advanced tumor stages is due to the presence of infiltrating LAIR1+ immune cells or LAIR1^+^ cancer cells. Accurate and detailed single-cell RNA sequencing may help in this characterization.

Another important gap of our knowledge is the interplay among classical immune checkpoint molecules such as PD1 and CTLA4 with LAIR1. This may be relevant in any kind of tumor, both waking up an already present specific antitumor immune response or increasing the efficiency of anti-cancer vaccines. The ongoing trial NCT05572684 using NC410 (the dimeric LAIR-2 protein fused to a human IgG1 Fc domain, see Chapter 4.1) in combination with the anti-PD1 pembrolizumab in MSS and low-microsatellite stable CRC or ovarian cancer may give some insights into the cooperation of conventional ICI and LAIR1 [227]. However, this trial does not analyze the molecular mechanisms underlying the possible cross-talk between PD1 and LAIR1. This point is essential to define whether the blocking of both PD1 and LAIR1 can be used in combination. In the TME, it is conceivable that PD1 interacts with PDL1 on tumor cells while LAIR1 binds to collagen in the stroma. This would suggest that effector immune cells could be strongly stimulated to respond due to a lack of tumor- and stroma-mediated inhibition. However, the biochemical mechanisms activated simultaneously by PD1 and LAIR1 when targeted with blocking antibodies should be studied in detail to tune these two potent immune regulatory checkpoints. On the other hand, the side effects of this blockade should be considered, as PD1 is involved in the adaptive immune response and LAIR1 can target the innate arm of the immune system as well. On the other hand, the efficiency of anticancer vaccines could be markedly increased by relieving APCs from LAIR1 inhibition. The study of the adaptive immune response to tumor-associated antigens or neoantigens is an open field of study. It is conceivable that an exhaustive analysis of the adaptive immune response and the role of LAIR1/LAIR2 in regulating the processing abilities of APCs is necessary to plan the targeting of this molecule to modulate specific T cell responses. Finally, the role of other non-conventional inhibitory molecules and LAIR1 should be studied. Overall, the promising information from preclinical and ongoing clinical trial seems to be the first step for planning the future targeting of LAIR1 isoforms and/or LAIR2. The levels of LAIR2 could be evaluated in both hematological and solid tumors to plan the targeting of LAIR1-mediated inhibitory effects. Further, LAIR2 might be studied as a biomarker of some leukemia therapy response or as a mechanism of resistance. Overall, a brief summary of some of these considerations is shown in Figure 5.

In this section, we do not consider the role of LAIR1 molecules in inflammation and/or autoimmune diseases and malaria, but these topics will be of great interest to designing LAIR-targeting therapeutics for these pathological conditions. In this case, it should be noted that the targeting of LAIR1 should elicit a negative signal in effector immune cells to switch down the undesired hyperreactivity and inflammation against self-cells.

## 10. Conclusions

Although the presence of LAIR1 and its inhibitory role on immune cells has been documented, its function during tumorigenesis and tumor progression remains contradictory. The widespread expression of LAIR1 and its ligands underscores the importance of its inhibitory function. In vivo, the deregulation of LAIR1 leads to an increase in uncontrolled inflammation, as well as its hyperregulation in some types of cancer, which has been associated with disease progression and may act as a predictor of clinical outcomes. In conclusion, there is clear evidence that LAIR1 represents in the oncologic field a very attractive therapeutic target for the treatment of hematologic malignancies for its strong expression confined to leukemic stem cells. This would imply that its targeting could lead to a cure for some hematological malignancies. On the other hand, LAIR1 is expressed on both antitumor immune cells and solid tumor cells, and for this reason it can deliver opposite effects in the TME. However, the blocking of the LAIR1-mediated signal in tumor-infiltrating lymphocytes should elicit a strong antitumor immune response in both T and NK cells. This could be the rationale to use the targeting of LAIR1 in solid tumor therapy as well. This scenario is further complicated by the possible presence of LAIR2, which competes with LAIR1, leading to the inhibition of the LAIR1-mediated signal. The use of novel drugs recently produced (NC525 and NC410) in a clinical setting should consider all of these aspects to avoid unwanted effects and hypothesize their use in combination with other immune checkpoint inhibitors. On the basis of the experimental evidence reported in this review, these tools could be applied mainly if not exclusively in some hematological malignancies.

## Figures and Tables

**Figure 1 biomolecules-15-00866-f001:**
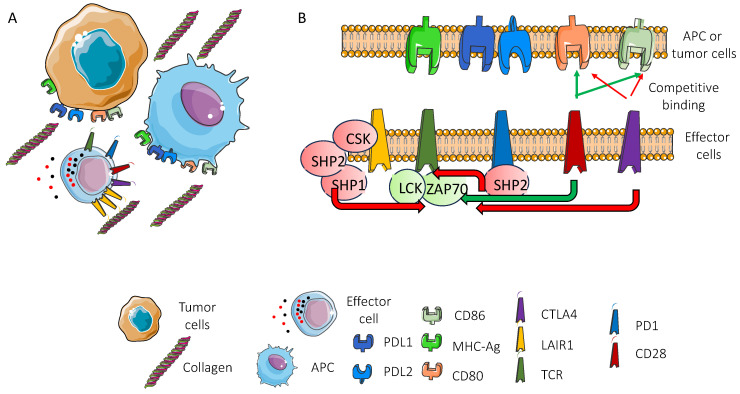
LAIR1-mediated molecular mechanisms of inhibition of immune cell function. (**A**) effector cells (NK or T lymphocytes) can interact with antigen-presenting cells (APCs, depicted in light blue) or tumor cells (brown cell) through different receptor–ligand pairs represented with different colors. Collagen, a main component of the extracellular matrix, is present among cells. LAIR1 (light brown receptor) is expressed on effector lymphocytes. As explained in the text, LAIR1 can deliver an inhibitory signal after interaction with collagen. The receptors expressed on lymphocytes such as TCR, CD28, PD1, and CTLA4 can deliver activating or inhibitory signals according to their association with transducer molecules. The expression of LAIR1 on APC or tumor cells is not shown in this figure to simplify the scenario. The figure is focused on TCR-mediated signal transduction in T lymphocytes. Here, the response of NK cells is not shown. (**B**) the interaction among the receptors on lymphocytes and ligands expressed on either APC or tumor cells has been magnified to show in detail the molecular interactions among the different molecules and the transducer molecules to which these receptors can be associated. The recognition of the antigen is mediated by the T cell receptor (depicted in dark green, first signal), which in turn activates the immune response. A second signal is delivered by the binding of CD28 (depicted in dark red) with CD80 or CD86 molecules. TCR and CD28 molecules concur in the activation of kinases such as LCK and/or ZAP70. The second co-stimulatory signal is needed to achieve the full activation of the effector cell, leading to lymphocyte proliferation, the killing of tumor target cells, or the production of cytokines. The regulatory receptor CTLA4 competes with CD28 for binding with CD80 or CD86, impairing the delivery of the second signal. PD1 interacting with PDL1 or PDL2 triggers the SHP2 phosphatase activity to block TCR signaling, impairing the activation of LCK and/or ZAP70 kinases. LAIR1 binds to collagen and associates with SHP1 or SHP2, similarly to PD1, which can reduce the phosphorylation of ZAP70 or LCK. Also, LAIR1 can engage the C-terminal Src kinase (CSK), regulating the receptor signal-mediated activation through tyrosine phosphorylation. The red lines indicate the inhibitory effect, while the green one indicates the activating effect. Templates of cell and molecule are from https://smart.servier.com/.

**Figure 2 biomolecules-15-00866-f002:**
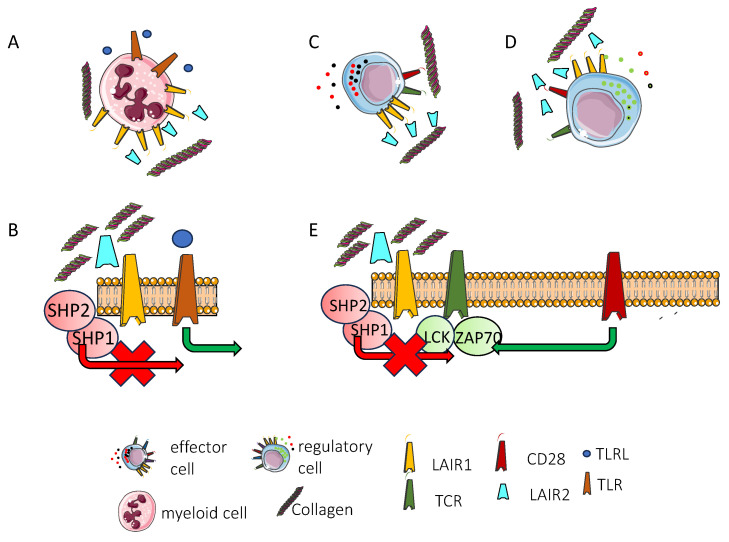
LAIR2 can regulate the signal delivered through LAIR1. (**A**) myeloid cells (pink) expressing the LAIR1 molecule (light brown) can be involved in inflammation processes, and LAIR2 (light blue), a soluble form of LAIR1, can be present in the microenvironment. LAIR2 has a greater affinity for collagen than LAIR1 and it can compete with LAIR1 for binding to collagen. The engagement of LAIR1 by collagen delivers an inhibitory signal. Competition with LAIR2 relieves the cells from this inhibitory signal. The net effect of this competition will depend on the level of expression of LAIR1, the concentration of LAIR2, and the presence of collagen in the microenvironment. (**B**) the collagen engagement of LAIR1 downregulates inflammation in myeloid cells. Indeed, the activating signal delivered through the Toll-like receptor (TLR) ligand (blue circle)–TLR (brown receptor) interaction is downregulated by SHP1/SHP2 phosphatases (red arrow). LAIR2, competing with LAIR1 for binding with collagen, blocks the negative signal (red cross) and allows the delivery of TLR-mediated activation (green arrow). (**C**) effector lymphocytes can be activated by recognition of the antigen. (**D**) regulatory cells inhibit several functions of effector cells by releasing immunomodulating cytokines. (**E**) the triggering of T cells through TCR (dark green) and CD28 (dark red) leading to the activation of LCK and ZAP70 is downregulated by LAIR1 binding with collagen (red arrow). LAIR2 inhibits (red cross) this signal by competition with LAIR1 allowing to TCR-mediated activation. It is conceivable that LAIR2 facilitates the activation of effector cells. Likewise, the inhibitory function of regulatory T cells can be elicited in the presence of LAIR2. Templates of cell and molecule are from https://smart.servier.com/.

**Figure 3 biomolecules-15-00866-f003:**
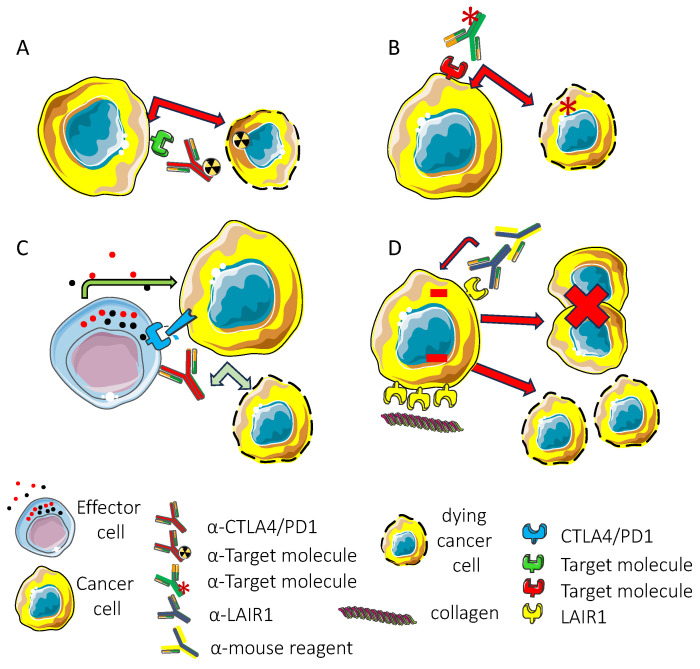
Targeting of tumor cells with antibodies to tumor-expressed molecules. Antibodies target tumor surface molecules, leading to the inhibition of tumor cell growth and dissemination. The antibodies can recognize “target molecules” highly expressed on tumor cells (green and red receptors). The antibody-mediated effect can be elicited through different molecular mechanisms. (**A**) the antibody conjugated to a radioisotope recognizes the target molecule, and after internalization in tumor target cells, the radioisotope induces cell death. (**B**) the antibody linked to a cytotoxic drug (usually inhibitors of microtubule poly or depolymerization) (*) interacts with the target molecule; after internalization, the drug is released and it blocks essential biological components of the tumor target cell involved in cell proliferation and/or the maintenance of cellular homeostasis; (**C**) antibodies to classical immune checkpoint receptor molecules such as CTLA4 and/or PD1 can impair the binding of the inhibitory receptor expressed on lymphocytes with the corresponding ligand on tumor cells. As a consequence, the inhibitory receptor does not deliver an inhibitory signal on the activation of effector cell functions triggered through activating surface receptors such as TCR. This can allow effector T cells to exert their antitumor activity by recognizing the tumor antigens, releasing cytotoxic molecules or antitumor cytokines. The result of this effect is the killing of the tumor cell. (**D**) The inhibitory molecule expressed on tumor target cells (such as LAIR1 on leukemic cells) upon its cross-linking achieved by interaction with the ligand or appropriate second reagents can lead to an inhibitory signal in tumor cells. This signal can impair cell proliferation and/or induce cell death. It should be noted that the anti-LAIR1 mAb can function similarly to anti-PD1 when expressed on effector cells, while the engagement of LAIR1 on tumor cells, typically in hematological malignancies, can directly evoke inhibitory signals in tumor cells. Experimentally, the cross-linking of LAIR1 is achieved using an anti-mouse reagent. In vivo, this cross-linking is triggered by collagen, which expresses repetitive binding sites allowing the engagement of multiple LAIR1 molecules. Red arrows indicate negative signals, while the green ones indicate positive signals. The red cross indicates the block of proliferation. This figure does not show the effect of therapeutic antibodies mediated by immune cells such as antibody-dependent cellular cytotoxicity or complement dependent cytotoxicity. Templates of cell and molecule are from https://smart.servier.com/.

**Figure 4 biomolecules-15-00866-f004:**
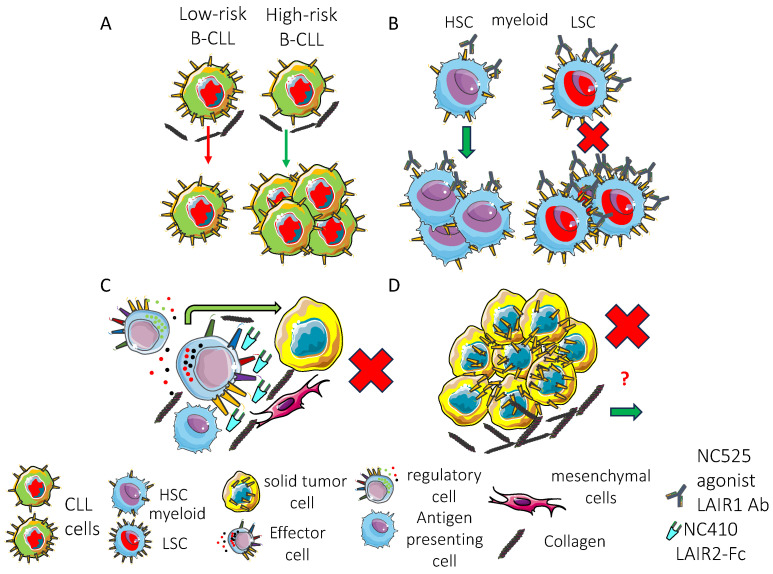
LAIR1 can play a role in the regulation of growth of some hematological malignancies, while in solid tumors its function is not well-defined. (**A**) LAIR1 expression levels and effects on CLLs. LAIR1 is expressed at different levels on the cell surface of B cells of chronic lymphocytic leukemia (CLL). Low-risk (LR) patients expressed high levels of LAIR1, conversely to high-risk (HR) patients. It is conceivable that LAIR1 engagement by collagen does not block the proliferation of HR CLLs (green arrow), while LR CLLs are controlled by LAIR1-mediated inhibitory signal (red arrow). (**B**) LAIR1 effects on myeloid leukemia. High levels of surface expression of LAIR1 on myeloid leukemia stem cells (LSCs) allow the delivery of an inhibitory signal through the LAIR1 agonist antibody (NC525) leading to LSC cell death and blocking leukemia proliferation (red cross). The NC525 antibody does not alter the growth of healthy hematopoietic stem cells (HSCs) because the LAIR1 level is too low to inhibit proliferation or induce cell death (green arrow). (**C**) therapeutic role of LAIR2. The LAIR2-Fc protein (NC410) can compete with surface LAIR1 for binding with collagen. This reduces/abolishes the LAIR1-mediated negative signal, allowing the activation of antitumor cells against tumor target cells. The net effect will depend on the presence of effector and/or regulatory cells in the tumor microenvironment expressing LAIR1. If effector cells prevail over regulatory cells, the tumor will not grow (red cross). (**D**) effects in solid tumors. LAIR1 expressed on solid tumors is associated with protumor or antitumor effects (see Table 3) leading to either tumor cell proliferation or tumor cell death. Deeper phenotypic and functional analyses are needed to properly define its role in solid tumor biology (red question mark). Templates of cell and molecule are from https://smart.servier.com/.

**Figure 5 biomolecules-15-00866-f005:**
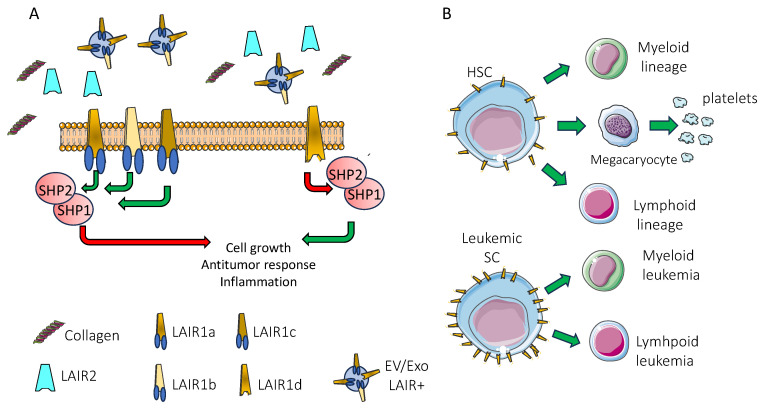
Focus on some of the possible studies to fill the gaps of knowledge on the function of LAIR1. (**A**) the expression and function of the LAIR1 isoforms (brown) can give some insights on the differential role of these molecules expressed on the several leukocytes subsets. It should be noted that the presence of the LAIR1d isoform (on the **right**) lacking the ITIM motifs can impair the signal mediated by the other LAIR isoforms bearing these ITIMs (on the **left**). The presence in the microenvironment of LAIR2 (light blue), collagen, and microvesicles/exosomes (EV/Exo LAIR) expressing isoforms of LAIR1 may directly influence the signal elicited upon the engagement of LAIR1 expressed at the cell surface. The binding of collagen with LAIR1a or LAIR1b or LAIR1c isoforms activates SHP1 and SHP2, blocking the function of the cells while binding with LAIR1d does not elicit the inhibitory signal, allowing cell growth, antitumor activities, and the triggering of inflammation. (**B**) a detailed analysis of the expression and effects of LAIR1 isoforms’ engagement on hematopoietic stem cells (HSCs) regulating differentiation of the myeloid and lymphoid lineages and platelets could help to better plan the targeting of leukemic stem cells (LSCs) to plan therapies to eliminate myeloid or lymphoid leukemias. Templates of cell and molecule are from https://smart.servier.com/.

**Table 1 biomolecules-15-00866-t001:** Antibodies and main applications.

Antibody Clone Name	Antibody Feature	Immunization with	Assays Applied	Preclinical/ Clinical Application	Main Reference
9.1C3	IgG2a mo anti-h LAIR1	LGL	FC, IHC, FA	ND/ND	[41]
NKTA255	IgG1 mo anti-h LAIR1	NK cells and CD2- thymocytes	FC, WB, FA	ND/ND	[42]
NKTA72	IgG1 mo anti-h LAIR1	NK cells and CD2- thymocytes	FC, WB, FA	ND/ND	[42]
1B1	IgG1 mo anti-h LAIR1	NK cell clones	FC, WB, FA	ND/ND	[42]
1F1	IgG1 mo anti-h LAIR1	NK cell clones	FC, WB, FA	ND/ND	[42]
DX26	IgG1 mo anti-h LAIR1	NK cells	FC, WB, FA	ND/ND	[43]
8A8	IgG1k mo anti-h LAIR1	Not determined	FA	ND/ND	[51]
lc12 (ab14826)	IgG mo anti-h LAIR1	Abcam, not defined	IF	Yes/ND	[52]
(F-5) sc-398141	IgG2a mo anti-h LAIR1	Amino acids 182–287 mapping at the C-terminus of LAIR1 of human origin	WB, IP, IF, ELISA	ND/ND	[53]
Antisera 145 Antisera 148	Polyclonal rabbit anti-h LAIR1	145 * 148 **	IP * WB **	ND/ND	[54]
3G4	Mo anti-h IgG1	LAIR2-GST fusion protein	WB/FA ^	ND/ND	[55]
HPA011155	Polyclonal rabbit anti-h LAIR1	Not defined	FC, WB, IHC	ND/ND	[56]
342219 FAB2664JF525	IgG2b mo anti-h LAIR1	rh LAIR1 isoform 1 Gln22-His163	WB, FC, CyTOF-ready	ND/ND	[57]
Monoclonal 113	IgG Armenian hamster anti-h	Abcam Not determined	WB, IP, FC, FA	ND/ND	[58]
NGM438	IgG1 humanized anti-h LAIR1	Not determined	High-affinity therapeutic mAb to antagonize LAIR1 in solid cancers	ND/ND	[59]
NGM438	Modified to react with mouse LAIR1	Not determined	Therapeutic antibody in mouse models	Yes/ND	[59]
NC525	IgG1k Humanized anti-h LAIR1	Not determined	Reactivity with LAIR1+ AML cells	Yes/Yes	[60]
FMU-mLAIR-1.1, FMU-mLAIR-1.2, FMU-mLAIR-1.3	IgM IgG1 IgM Rat anti-mouse LAIR1	Murine LAIR1-Fc	Reactivity with murine LAIR1 cells IHC, WB, FC	Possible use in murine models	[61]

IF: immunofluorescence; FC: flow cytometry; IHC: immunohistochemistry; FA: functional assay; IP: immunoprecipitation; WB: Western blot; ELISA: enzyme-linked immunosorbent assay; mo anti-h: mouse anti-human; NK: natural killer; LGL: large granular lymphocyte; AML: acute myeloid leukemia; * produced against the intracellular domain of LAIR1; ** produced against the extracellular domain of LAIR1; ^ this antibody inhibits binding to collagen but not cytotoxicity; ND: not done in the reference cited; Yes: used in preclinical models.

**Table 2 biomolecules-15-00866-t002:** Major inhibitory receptors expressed on immune cells.

Receptors Designation Cluster Differentiation	Ligand	Cell Expression	Ligand Expression	Reference
LAIR1/CD305 LAIR2/CD306	Collagen, C1q, adiponectin, surfactant protein D, RIFIN	T, B, NK, myeloid cells, tumor cells	Every tissue	[96]
Inhibitory KIR/CD158	MHC-I allele	NK, T cells	Almost all nucleated cells	[97]
Activating KIR/CD158	MHC-I allele	NK, T cells	Almost all nucleated cells	[97]
Inhibitory CLIR CD94/NKG2B CD94/NKG2A	MHC-I allele	NK, T cells	Almost all nucleated cells	[71]
Activating CLIR CD94/NKG2C	MHC-I allele	NK, T cells	Almost all nucleated cells	[98]
ILT/LILRB/LIR/CD85	Several HLA-I antigens, HLA-G	Myeloid cells, B cells	Almost all nucleated cells	[99,100]
Irp60/CD300a	PE, PS	Myeloid cells, B cells	activated, infected, transformed, or apoptotic cells	[101]
IREM1/CD300f	Not determined in humans, norovirus receptor in mice	Myeloid cells and mast cells		[102]
Siglec1-13, 15-17 inhibiting	Glycans with sialic acid	Myeloid cells, B cells, osteoclasts	Different cell types	[103]
Siglec14 activating	Glycans with sialic acid	Primary and secondary lymphoid organs, subsets of innate cells	Different cell types	[103]
CTLA4/CD152	CD80/CD86	Activated T cells several subsets (Treg), tumor cells of different histotypes	Antigen-presenting cells	[26]
PD1/CD274	PDL1/PDL2	T, NK cells	Antigen-presenting cells, several tumor cells	[104]
TIM3/CD366/HAVCR2	Hepatitis virus A cellular receptor 2, Galectin 9, phosphatidyl serine, HMGB1, CEACAM1	T, NK cells	Different cell types, antigen-presenting cells, epithelial cells	[105]
TIGIT/ VSIG9	CD155/PVR, CD112/PVRL2	T, NK cells	High expression on tumor cells, low expression in normal cells	[106]
LAG3/CD223	MHC-II, Galectin 3, FGL1	T cells, NK cells, pDC B cells	Antigen-presenting cells, activated lymphocytes	[107]
CD96/TACTILE	CD155/PVR	NK cells, T cells	High expression on tumor cells, low expression in normal cells	[108]
VISTA	VSIG-3, PSGL-1, VISIG-8, Galectin 9, Sdc-2, LRIG-1	T subsets, Treg, MDSC	Several types of cells, immune cells	[109]

RIFIN: repetitive interspersed family of polypeptides; KIR: killer immunoglobulin-like receptor; MHC: major histocompatibility complex; CLIR: C lectin type inhibitory receptor; NKG2: NK group 2; LILR: leukocyte immunoglobulin-like receptor subfamily; ILT: inhibitory lymphocyte transcript; LIR: leukocyte Ig-like receptor; Irp: inhibitory receptor protein; PE: phosphoethanolamine; PS: phosphatidylserine; HLA: human leukocyte antigen; IREM: immune receptor expressed on myeloid cells; Siglec: sialic acid-binding immunoglobulin-type lectins; CTLA: cytotoxic T lymphocyte antigen; Treg: regulatory T cells; PD: programmed death receptor; PDL: programmed death receptor ligand; HAVCR: hepatitis A virus cellular receptor; TIM: T-cell immunoglobulin and mucin-domain containing; HMGB: high-motility group box; CEACAM: carcinoembryonic antigen-related cell adhesion molecule; TIGIT: T cell immunoreceptor with Ig and ITIM domains; PVR: polio virus receptor; VISIG: V-set and immunoglobulin domain containing; LAG: lymphocyte activation gene; FGL: fibrinogen-like protein; TACTILE: T cell activation, increased late expression; VISTA: V-domain Ig suppressor of T cell activation; PSGL: P selectin glycoprotein ligand; Sdc: syndecan; LRIG: leucine-rich repeats and immunoglobulin-like domains protein; MDSC: myeloid-derived suppressor cells; NK: natural killer; pDC: plasmocytoid dendritic cell.

**Table 3 biomolecules-15-00866-t003:** Expression and function of LAIR1 on solid tumor cells.

Solid Tumor Type	LAIR1 Expression	LAIR1-Mediated Function After Silencing	LAIR1-Mediated Function After Overexpression	LAIR1 Association with Biological Features of Solid Tumor	Reference DOI
Ovarian carcinoma	Tumor tissues Some cell lines	Increase in proliferation, colony formation, matrix invasion	Inhibition of proliferation, migration, induction apoptosis	Correlates with grade	[52,193]
Breast carcinoma	Tumor tissues, some cell lines	Reduces proliferation and invasion	ND	Correlates with shorter patient survival, and grade	[194]
Renal cell carcinoma	Tumor tissues	Reduces proliferation	Increase in proliferation	Correlates with shorter survival	[56]
Cervical carcinoma	Tumor tissues	ND	Reduction in proliferation and anti-apoptosis capacity	Correlates with the grade	[199]
Osteosarcoma	Tumor tissues and cell lines and healthy osteoblasts	ND	Inhibition of migration and EMT	Correlates with the stage	[53]
Glioblastoma	Low-grade glioma, some cell lines	ND	In murine models, large tumor with LAIR1 overexpression		[200]
Hepatocellular carcinoma	Tumor tissues, association with low differentiation	Increasing PDL1 expression		Worse OS	[196,201]
Oral squamous cell carcinoma	Tumor tissue, mainly associated with leukocyte infiltration *	ND	ND	Association with grade and immunosuppressive cells (MDSC, M2)	[195]

* LAIR1^+^ cells associated with leukocyte infiltration. No double IHC staining to define the cells expressing LAIR1, not clearly expressed on tumor cells. Red color: negative effect; green color: positive effect. ND: not determined; EMT: epithelial-mesenchymal transition; PDL1: programmed cell death receptor ligand 1; OS: overall survival; MDSC: myeloid derived suppressor cells, M2: type 2 macrophage.

**Table 4 biomolecules-15-00866-t004:** Features of sequences of the different LAIR1 isoforms. LAIR1d does not express ITIM and does not deliver an inhibitory signal into the cell.

Gene/Agent	Isoform/Molecule	RefSeq (mRNA/protein)	Functional Role	Therapeutic Relevance	Clinical/Preclinical Tools	PMID/Trial
**LAIR1**	Isoform A (canonical)	NM_002287.4/NP_002278.2	Full-length inhibitory receptor with ITIM motifs	Immune checkpoint in tumors, infections, inflammation	Antagonist mAbs – preclinical	PMID: 38648067 PMID: 36211388
	MSPHPTALLGLVLCLAQTIHTQE**E**DLPR**PSISAEPGTVIPLGSHVTFVCRGPVGVQTFRLERDSRSTYNDTEDVSQASPSESEARFRIDSVREGNAGLYRCIYYKPPKWSEQSDYLE**LLVK**ESSGGPDSPDTEPGSSA**GPTQRPSDNSHNEHAPASQGLKAEHLYI **LIGVSVVFLFCLLLLVLFCL**HRQNQIKQGPP**RSKDEEQKPQQ**RPDLAVDVLERTADKATVNGLPEKDRETDTSALAAGSSQE**VTYAQL**DHWALTQRTARAVSPQSTKPMAES**ITYAAV**ARH					
**LAIR1**	Isoform B	NM_001289025.3/NP_001275954.2	C-terminal variant	Unknown	None known	—
	MSPHPTALLGLVLCLAQTIHTQE**E**DLPR**PSISAEPGTVIPLGSHVTFVCRGPVGVQTFRLERDSRSTYNDTEDVSQASPSESEARFRIDSVREGNAGLYRCIYYKPPKWSEQSDYLE**LLVK**XXXXXXXXXXXXXXXXX**GPTQRPSDNSHNEHAPASQGLKAEHLYI **LIGVSVVFLFCLLLLVLFCL**HRQNQIKQGPP**RSKDEEQKPQQ**RPDLAVDVLERTADKATVNGLPEKDRETDTSALAAGSSQE**VTYAQL**DHWALTQRTARAVSPQSTKPMAES**ITYAAV**ARH					
**LAIR1**	Isoform C	NM_001289026.3/NP_001275955.2	Truncated cytoplasmic tail	Not established	None known	—
	MSPHPTALLGLVLCLAQTIHTQE**X**DLPR**PSISAEPGTVIPLGSHVTFVCRGPVGVQTFRLERDSRSTYNDTEDVSQASPSESEARFRIDSVREGNAGLYRCIYYKPPKWSEQSDYLE**LLVK**XXXXXXXXXXXXXXXXX**GPTQRPSDNSHNEHAPASQGLKAEHLYI **LIGVSVVFLFCLLLLVLFCL**HRQNQIKQGPP**RSKDEEQKPQQ**RPDLAVDVLERTADKATVNGLPEKDRETDTSALAAGSSQE**VTYAQL**DHWALTQRTARAVSPQSTKPMAES**ITYAAV**ARH					
**LAIR1**	Isoform D	NM_001289027.3/NP_001275956.2	Lacks ITIMs; potential decoy	No validated role	None known	—
	MSPHPTALLGLVLCLAQTIHTQE**E**DLPR**PSISAEPGTVIPLGSHVTFVCRGPVGVQTFRLERDSRSTYNDTEDVSQASPSESEARFRIDSVREGNAGLYRCIYYKPPKWSEQSDYLE**LLVK**ESSGGPDSPDTEPGSSA**GPTQRPSDNSHNEHAPASQGLKAEHLYI **LIGVSVVFLFCLLLLVLFCL**HRQNQIKQGPP**RSKDEEQKPQQ**R					
**LAIR2**	Soluble isoform	NM_001014835.2/NP_001014835.1	Secreted decoy receptor that binds collagen and C1q	Potential immune modulator and biomarker	**NC410 (LAIR-2–Fc)–Phase I clinical trial**	PMID: 34121658 PMID:38236251 NCT04408599 NCT05572684 NCT06941857
	MSPHLTALLGLVLCLAQTIHTQEGALPR**PSISAEPGTVISPGSHVTFMCRGPVGVQTFRLEREDRAKYKDSYNVFRLGPSESEARFHIDSVSEGNAGLYRCLYYKPPGWSEHSDFLE**LLVK**ESSGGPDSPDTEPGSSA**GTVPGTEASGFDAP					

Sequential alignments: red: lacking residues; blue: basic and acid residues; cyan highlighted: polar residues; purple: Ig-like C2-type domain; green: transmembrane domain; yellow highlighted: ITIM motifs 1 and 2.

## Data Availability

Not applicable.

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
