# Peer review of "The Role of LAIR1 as a Regulatory Receptor of Antitumor Immune Cell Responses and Tumor Cell Growth and Expansion"

_biomolecules, 2025, doi:10.3390/biom15060866_

Round 1

Reviewer 1 Report

Comments and Suggestions for Authors

1. Summary

The review article by Poggi et al. explores the role of leukocyte-associated immunoglobulin-like receptor 1 (LAIR1) as a key inhibitory molecule regulating immune cell functions and tumor cell growth. The manuscript provides a comprehensive overview of the molecular characteristics of LAIR1, its signaling pathways, expression patterns across different immune cell types, potential involvement in disease processes, and the therapeutic implications of targeting LAIR1. Strengths of the paper include a thorough historical background, insightful molecular comparisons with other inhibitory receptors, and a detailed discussion of LAIR1’s function across various physiological and pathological settings.

2. General Comments

The manuscript is well-organized, comprehensive, and demonstrates a strong grasp of the literature. It offers an extensive summary of previous and current work on LAIR1 in immune regulation and oncology.

While the work is commendable in its breadth, its impact would be greatly enhanced by a more focused identification of critical knowledge gaps and a more systematic outline of future research directions. In several sections, heavy molecular and historical detail somewhat obscures the broader conceptual link to current challenges in the field and potential clinical applications.

In addition, from a hematological perspective, several important aspects warrant further attention. Although LAIR1 is widely expressed among immune cells, the manuscript does not sufficiently emphasize its predominant expression and regulatory role within the hematopoietic system. The dynamic changes of LAIR1 expression across different stages of hematopoietic differentiation, and its significance in hematopoietic homeostasis, are not fully explored. Furthermore, although the review briefly touches on LAIR1 expression in hematological malignancies such as CLL and AML, the discussion remains superficial and does not delve into potential associations with disease subtypes, prognosis, immune evasion, or therapeutic resistance. The therapeutic discussion is disproportionately focused on solid tumors, overlooking the unique relevance of LAIR1 as a potential target in blood cancers. Additionally, the role of soluble LAIR2 as a biomarker or therapeutic modulator in hematologic disease is underexplored.

To strengthen the manuscript, I encourage the authors to include a dedicated section on LAIR1’s role in the hematopoietic system, provide a deeper analysis of its relevance in hematological malignancies, and discuss opportunities for therapeutic targeting of LAIR1 and LAIR2. Revising the conclusion to emphasize the hematopoietic context as a major arena for LAIR1 research would also enhance the overall coherence and clinical significance of the paper.

3. Article Comments

Hypotheses:

While the absence of specific hypotheses is appropriate for a review article, the authors could strengthen the manuscript by suggesting explicit future research directions or testable hypotheses based on current gaps in the field.

Methodological Comments:

Although no original experiments are described, when summarizing existing literature, it would be helpful to briefly mention important methodological limitations where relevant (e.g., species-specific differences between human and murine LAIR1 models).

4. Review-Specific Comments

Completeness and Relevance:

The manuscript is generally complete and covers the major aspects of LAIR1 biology. The topic is highly relevant to the field of immune regulation and cancer immunotherapy.

Identification of Knowledge Gaps:

The review would benefit from a clearer articulation of major unanswered questions, particularly regarding

  1. The mechanisms underlying LAIR1’s dual roles in inflammation and immune suppression.

  2. The specific roles of LAIR2 in physiological versus pathological states.

Citations:

The citations are mostly appropriate and up-to-date, with some essential older references appropriately included. Self-citation is moderate and acceptable.

Coherence of Statements and Logical Flow:

Some conceptual connections could be better integrated, particularly the interplay between LAIR1 and other immune checkpoint pathways such as PD-1.

Figures and Tables:

  • Tables I and II effectively summarize information and are valuable references.

  • Figures 1 and 2 are conceptually important but could benefit from improved clarity, particularly by reducing overcrowding and improving separation of schematic elements.

  • Figure 3 provides a useful overview of therapeutic strategies but would be clearer with simplified graphics and expanded legends to enhance accessibility for a broader audience.

5. Specific Comments

  • Lines 89–90: Typographical error: “additkional” should be corrected to “additional.”

  • Table I: Consider adding a column indicating whether each antibody has been tested in clinical or preclinical models.

  • Section 2.2 (Molecular Features): Further clarification on potential functional differences among LAIR1 isoforms would enhance this section.

  • Section 3.3 (APCs): The discussion is strong but could be improved by linking LAIR1 regulation of APCs to implications for vaccine responses or tumor immunotherapy strategies.

  • Figure 1: The inset panels are conceptually helpful, but the visual layout could be improved for easier interpretation.

  • Figures 2 and 3: Figures are informative but appear overcrowded; consider simplifying or breaking them into smaller, clearer panels.

Author Response

Reviewer 1

  1. Summary

The review article by Poggi et al. explores the role of leukocyte-associated immunoglobulin-like receptor 1 (LAIR1) as a key inhibitory molecule regulating immune cell functions and tumor cell growth. The manuscript provides a comprehensive overview of the molecular characteristics of LAIR1, its signaling pathways, expression patterns across different immune cell types, potential involvement in disease processes, and the therapeutic implications of targeting LAIR1. Strengths of the paper include a thorough historical background, insightful molecular comparisons with other inhibitory receptors, and a detailed discussion of LAIR1’s function across various physiological and pathological settings.

  1. General Comments

The manuscript is well-organized, comprehensive, and demonstrates a strong grasp of the literature. It offers an extensive summary of previous and current work on LAIR1 in immune regulation and oncology.

While the work is commendable in its breadth, its impact would be greatly enhanced by a more focused identification of critical knowledge gaps and a more systematic outline of future research directions. In several sections, heavy molecular and historical detail somewhat obscures the broader conceptual link to current challenges in the field and potential clinical applications.

In addition, from a hematological perspective, several important aspects warrant further attention. Although LAIR1 is widely expressed among immune cells, the manuscript does not sufficiently emphasize its predominant expression and regulatory role within the hematopoietic system. The dynamic changes of LAIR1 expression across different stages of hematopoietic differentiation, and its significance in hematopoietic homeostasis, are not fully explored. Furthermore, although the review briefly touches on LAIR1 expression in hematological malignancies such as CLL and AML, the discussion remains superficial and does not delve into potential associations with disease subtypes, prognosis, immune evasion, or therapeutic resistance. The therapeutic discussion is disproportionately focused on solid tumors, overlooking the unique relevance of LAIR1 as a potential target in blood cancers. Additionally, the role of soluble LAIR2 as a biomarker or therapeutic modulator in hematologic disease is underexplored.

To strengthen the manuscript, I encourage the authors to include a dedicated section on LAIR1’s role in the hematopoietic system, provide a deeper analysis of its relevance in hematological malignancies, and discuss opportunities for therapeutic targeting of LAIR1 and LAIR2. Revising the conclusion to emphasize the hematopoietic context as a major arena for LAIR1 research would also enhance the overall coherence and clinical significance of the paper.

We have modified the review with some focusing on the relevance of LAIR1 on hematopoietic precursors as suggested. This part has been added lines 1550-1571, by adding a new specific chapter (chapter 8 entitled ” LAIR1 expression on hematopoietic cell precursors and role in the regulation of hematopoiesis and cell differentiation.”) to focus the attention of the reader on this specific topic. Also, the future therapeutic applications of tools interfering with LAIR1 and LAIR2 function have been stressed in the last new chapter 9. We added an additional table IV reporting some molecular features of the different isoforms of LAIR1 and an additional figure (Figure 5) to summarize the information reported.

We do not alter the flow of the information of the review as the topic of hematological malignancies was considered on pages 20-21-22 lines 818-971 but we further analyze the relevance of LAIR1 in the new chapter entitled” LAIR1 expression on hematopoietic cell precursors and role in the regulation of hematopoiesis and cell differentiation.

  1. Article Comments

Hypotheses:

While the absence of specific hypotheses is appropriate for a review article, the authors could strengthen the manuscript by suggesting explicit future research directions or testable hypotheses based on current gaps in the field.

Possible future directions have been added at the end of the manuscript in the chapter 9 ” Future research to identify the knowledge gaps on LAIR1 function and its therapeutic targeting.”

Methodological Comments:

Although no original experiments are described, when summarizing existing literature, it would be helpful to briefly mention important methodological limitations where relevant (e.g., species-specific differences between human and murine LAIR1 models).

Some consideration on this point has been added in the chapter

  1. Review-Specific Comments

Completeness and Relevance:

The manuscript is generally complete and covers the major aspects of LAIR1 biology. The topic is highly relevant to the field of immune regulation and cancer immunotherapy.

Identification of Knowledge Gaps:

The review would benefit from a clearer articulation of major unanswered questions, particularly regarding

  1. The mechanisms underlying LAIR1’s dual roles in inflammation and immune suppression.
  2. The specific roles of LAIR2 in physiological versus pathological states.

We considered these two points on page 27, line 1189 in a new chapter 9 entitled:” Future research to identify the knowledge gaps on LAIR1 function and its therapeutic targeting.”

Citations:

The citations are mostly appropriate and up-to-date, with some essential older references appropriately included. Self-citation is moderate and acceptable.

Coherence of Statements and Logical Flow:

Some conceptual connections could be better integrated, particularly the interplay between LAIR1 and other immune checkpoint pathways such as PD-1.

This interplay has been further discussed in the chapter 9 entitled: “Future research to identify the knowledge gaps on LAIR1 function and its therapeutic targeting.” At pages 29-31 line 1574-1665.

Figures and Tables:

  • Tables I and II effectively summarize information and are valuable references.
  • Figures 1 and 2 are conceptually important but could benefit from improved clarity, particularly by reducing overcrowding and improving separation of schematic elements. We modified the figures reducing the panels and eliminating the inset, explaining the content better in the corresponding legend. In the figure 2, we deleted the additional regulating molecules, such as CTLA4 and PD1 that were depicted in figure 1, to reduce the overcrowding and improve the readability of the figure.
  • Figure 3 provides a useful overview of therapeutic strategies but would be clearer with simplified graphics and expanded legends to enhance accessibility for a broader audience. We modified the figure accordingly explaining better the content in the corresponding legend. In particular, we separated the schemes regarding the different modalities of using antibodies to influence the tumor cell growth. We think this separation is useful to better understand the message of this figure.
  1. Specific Comments
  • Lines 89–90: Typographical error: “additkional” should be corrected to “additional.”
  • We corrected this misprint.
  • Table I: Consider adding a column indicating whether each antibody has been tested in clinical or preclinical models. We added a column with this information related to the article considered as reference for each antibody. In other words, we indicated whether in the paper cited the antibody has been used in preclinical or clinical settings.
  • Section 2.2 (Molecular Features): Further clarification on potential functional differences among LAIR1 isoforms would enhance this section. We added some molecular information about the different LAIR1 isoforms in the indicated section (lines 158-170 on page 5) as follows: In detail, the finding that anti-LAIR1 rabbit polyclonal antibody identified two different proteins in the myeloid cell line HL60 (45kD) and Jurkat T cell line (40kD) would suggest that different degrees of glycosylation as protein or alternative splicing as mRNA may exist of LAIR1 [56new] (doi: 10.1074/jbc.M001313200]. Indeed, molecular cloning from Jurkat T cells identified LAIR1c and LAIR1b while LAIR1a, LAIR1b and LAIR1d were from HL60. The LAIRc differs from LAIR1a for just an aminoacid while the LAIR1d lacks the ITIM intracellular domains. Importantly, LAIR-1d lacking ITIM may have a dominant negative role in the signaling mediated by ITIM-bearing isoforms of LAIR-1. Several other information on molecular features of LAIR1 are reported at https://www.ncbi.nlm.nih.gov/gene?Db=gene&Cmd=DetailsSearch&Term=3903, https://www.ensembl.org/Homo_sapiens/Gene/Summary?g=ENSG00000167613;r=19:54351384-54370558, and https://omim.org/entry/602992 .
  • Section 3.3 (APCs): The discussion is strong but could be improved by linking LAIR1 regulation of APCs to implications for vaccine responses or tumor immunotherapy strategies. This is a good point to be considered. We discussed the relevance of LAIR1 in APC regulation with the generation of vaccines at the end of the manuscript briefly in the new chapter 9.
  • Figure 1: The inset panels are conceptually helpful, but the visual layout could be improved for easier interpretation. Please see the replies to section “Figures and Tables” on this point.
  • Figures 2 and 3: Figures are informative but appear overcrowded; consider simplifying or breaking them into smaller, clearer panels. Please see the replies to section “Figures and Tables” on this point.

Reviewer 2 Report

Comments and Suggestions for Authors
  1. A systematic comparative analysis of LAIR1's functional roles and expression patterns across various diseases remains limited. Future studies should comprehensively compare LAIR1 expression profiles, molecular mechanisms, and therapeutic implications between different autoimmune diseases orcancers.
    2. Further investigations are warranted to elucidate the LAIR1-mediated crosstalk between tumor and immune cells, which may contribute to immune evasion and the establishment of immunosuppressive microenvironments. Considering the complexity of LAIR1's dual roles in immune regulation or tumor progress, integrating graphical summaries with empirical data could help resolve seemingly contradictory findings.
  2. Additionally, schematic diagrams that summarize LAIR1's cell-type-specific expression and signaling pathways in solid tumors would significantly enhance mechanistic understanding.

Author Response

Reviewer 2

  1. A systematic comparative analysis of LAIR1's functional roles and expression patterns across various diseases remains limited. Future studies should comprehensively compare LAIR1 expression profiles, molecular mechanisms, and therapeutic implications between different autoimmune diseases or cancers.

We agree with the reviewer that the analysis in several diseases is at the beginning regarding the role of LAIR1. We added some considerations on this point at the end of the review (lines 1661-1666). This review Is focused on LAIR1 in the oncology field as part of a specific special issue on this topic. Anyway, we cited and discussed the role of LAIR1 in different autoimmune disease but the focus is not inflammation or autoimmunity.

  1. Further investigations are warranted to elucidate the LAIR1-mediated crosstalk between tumor and immune cells, which may contribute to immune evasion and the establishment of immunosuppressive microenvironments. Considering the complexity of LAIR1's dual roles in immune regulation or tumor progress, integrating graphical summaries with empirical data could help resolve seemingly contradictory findings.

The role of LAIR1 appears to be to mediate a negative signal in leukocytes. This negative signal will inhibit the specific function of the immune cells. This has been deeply analyzed in this review.

The role of LAIR1 on hematological tumor follows the rule that LAIR1 gives a negative signal in target cells and by consequence proliferation of tumor cells is inhibited as reported in the papers cited regarding this topic.

The role of LAIR1 on solid tumor is difficult to be defined. Several experiments have been performed with reagents and settings different from those used for leukocytes. In several instances the surface expression of LAIR1 is questionable while intracellular expression is reported.

To our knowledge no experiments have been performed with the classical reagents used for leukocytes and some of the effects on function of solid tumor cells are reported after the artificially induced overexpression of LAIR1. In our opinion, it is too early to define the role of LAIR1 in solid tumor, also regarding possible therapeutic target of this molecule.

Respectfully, we consider to have analyzed several aspects of the LAIR1 in oncologic field as this manuscript has been submitted to a specific Special Issue on “"Advances in Cellular and Molecular Mechanisms in Immuno-Oncology and Onco-Hematology”. Of course, LAIR1 and LAIR2 can have a role in several other diseases (inflammatory, autoimmune) and healthy conditions (pregnancy) but the large majority of reports focused on immune cells and oncology. For this reason, we just mention in different parts of this review the reports outside from oncology field, instead of making a detailed analysis of them. Nevertheless, the addition of a new table to summarize data on solid tumor could help the reader to understand the role of LAIR in the topic dedicated by the special issue. Further, we changed the original title by adding the word “antitumor” to emphasize the specific analysis of role of LAIR1 in the oncologic field. 

  1. Additionally, schematic diagrams that summarize LAIR1's cell-type-specific expression and signaling pathways in solid tumors would significantly enhance mechanistic understanding.

An additional table (Table III) has been added to summarize data regarding the expression of LAIR1 on solid tumor cells (as suggested by reviewer 3).

Reviewer 3 Report

Comments and Suggestions for Authors

This review provides an exhaustive overview of the role of LAIR1 in the function of immune cells and cancer cells. Sections and figures explained very well the biology and the mechanism of action of LAIR1.

I suggest to the authors to include the following information to improve the quality of the manuscript

1) Please include a table describing LAIR1 expression and function on solid tumor cells as reported in paragraph 5.2. It would be helpful to the reader to understand the level of expression of LAIR1 on each cell line (if low or high), its role in each type of tumor described in the paragraph and the potential treatments targeting LAIR1 in each tumor

2) In paragraph 6 (LAIR1 expression and function in the tumor microenvironment) could author include additional information on the role of LAIR1 in immunosuppressive cells (i.e. Tregs, MDSCs, TAMs, CAFs etc)? Is LAIR1 expressed on these cells? If yes what is his role? Is LAIR1 involved in the recruitment of immunosuppressive cells in the TME?

Author Response

Reviewer 3

This review provides an exhaustive overview of the role of LAIR1 in the function of immune cells and cancer cells. Sections and figures explained very well the biology and the mechanism of action of LAIR1.

I suggest to the authors to include the following information to improve the quality of the manuscript

Please include a table describing LAIR1 expression and function on solid tumor cells as reported in paragraph 5.2. It would be helpful to the reader to understand the level of expression of LAIR1 on each cell line (if low or high), its role in each type of tumor described in the paragraph and the potential treatments targeting LAIR1 in each tumor

We have inserted this table as requested. Please see at page 24.   

2) In paragraph 6 (LAIR1 expression and function in the tumor microenvironment) could author include additional information on the role of LAIR1 in immunosuppressive cells (i.e. Tregs, MDSCs, TAMs, CAFs etc)? Is LAIR1 expressed on these cells? If yes what is his role? Is LAIR1 involved in the recruitment of immunosuppressive cells in the TME?

We have inserted some considerations on this topic as requested by the reviewer. Please see at page 26 lines 1418-1427 for information on the role of LAIR1 in migration and expression on regulatory cells; On the other hand, some information on the expression of LAIR1 on fibroblast like synoviocytes and the expression in the TME in association with stroma were already present in the original version in the chapter 6.

Reviewer 4 Report

Comments and Suggestions for Authors

The review article of Alessandro Poggi and colleagues focus on the leukocyte-associated immunoglobulin (Ig)-like receptor 1 (LAIR1). It belongs to the family of immune-inhibitory receptors and is widely expressed on almost all leukocytes as well as tumor cells. Four different types of ligands of LAIR1 have been described, including collagens, suggesting a potential immune-regulatory function on the extracellular matrix. By modulating cytokine secretion and cellular functions, LAIR1 displays distinct patterns of expression among leukocyte subsets during their differentiation and cellular activation and plays a major negative immunoregulatory role. It bears two immunoregulatory tyrosine-based inhibitory motif (ITIM) in the intracitoplasmatic protein domain, involved in the down regulation of signals mediated by activating receptors.

In this review, the authors  clearly summarize and discuss the role of LAIR1 in normal physiological conditions, as well as during pathological situations, including hematological and solid malignancies. Moreover the presence of both forms  cellular and soluble would indicate a fine regulation of the immunoregulatory activity. This scenario is further complicated by the possible presence of LAIR2, a soluble molecule which competes with LAIR1, leading to the inhibition of the LAIR1-mediated signal.

The authors conclude that the widespread expression of LAIR1 and its ligands underscores the importance of its inhibitory function. In vivo, deregulation of LAIR1 leads to an increase and uncontrolled inflammation, as well as its hyperregulation in some types of cancer, which has been associated with disease progression and may act as a predictor of clinical outcomes. There is clear evidence that LAIR1 represents a very attractive therapeutic target for the treatment of a variety of inflammatory and autoimmune diseases, as well as of hematologic malignancies. LAIR1 is expressed both on antitumor immune cells and tumor cells, and for this reason it can deliver opposite effects in the tumor microenvironment. The use of the novel drugs recently produced  in a clinical setting should consider all these aspects to avoid un-wanted effects and hypothesize their use in combo with other immune-checkpoint inhibitors.

The review is interesting and includes a balanced, comprehensive and critical view of the research area. It is well written and easy to read. Figures help the reader to better follow the text. Blocking LAIR1 signaling in immune cells represents a promising strategy for development of anti-cancer immunotherapy.

Minor points:

Lines 27-30. Please, re-write, perhaps a verb is missing.

REFERENCES: Control the font of some papers

Author Response

Reviewer 4

The review article of Alessandro Poggi and colleagues focus on the leukocyte-associated immunoglobulin (Ig)-like receptor 1 (LAIR1). It belongs to the family of immune-inhibitory receptors and is widely expressed on almost all leukocytes as well as tumor cells. Four different types of ligands of LAIR1 have been described, including collagens, suggesting a potential immune-regulatory function on the extracellular matrix. By modulating cytokine secretion and cellular functions, LAIR1 displays distinct patterns of expression among leukocyte subsets during their differentiation and cellular activation and plays a major negative immunoregulatory role. It bears two immunoregulatory tyrosine-based inhibitory motif (ITIM) in the intracitoplasmatic protein domain, involved in the down regulation of signals mediated by activating receptors.

In this review, the authors clearly summarize and discuss the role of LAIR1 in normal physiological conditions, as well as during pathological situations, including hematological and solid malignancies. Moreover the presence of both forms cellular and soluble would indicate a fine regulation of the immunoregulatory activity. This scenario is further complicated by the possible presence of LAIR2, a soluble molecule which competes with LAIR1, leading to the inhibition of the LAIR1-mediated signal.

The authors conclude that the widespread expression of LAIR1 and its ligands underscores the importance of its inhibitory function. In vivo, deregulation of LAIR1 leads to an increase and uncontrolled inflammation, as well as its hyperregulation in some types of cancer, which has been associated with disease progression and may act as a predictor of clinical outcomes. There is clear evidence that LAIR1 represents a very attractive therapeutic target for the treatment of a variety of inflammatory and autoimmune diseases, as well as of hematologic malignancies. LAIR1 is expressed both on antitumor immune cells and tumor cells, and for this reason it can deliver opposite effects in the tumor microenvironment. The use of the novel drugs recently produced in a clinical setting should consider all these aspects to avoid un-wanted effects and hypothesize their use in combo with other immune-checkpoint inhibitors.

The review is interesting and includes a balanced, comprehensive and critical view of the research area. It is well written and easy to read. Figures help the reader to better follow the text. Blocking LAIR1 signaling in immune cells represents a promising strategy for development of anti-cancer immunotherapy.

Minor points:

Lines 27-30. Please, re-write, perhaps a verb is missing. We corrected the sentence.

REFERENCES: Control the font of some papers We checked the font and the space among lines of the reference list.

Round 2

Reviewer 1 Report

Comments and Suggestions for Authors

Dear Authors,

Thank you for your thorough revision. The new material really brings out LAIR1’s importance in normal and malignant blood cells. A few friendly suggestions to polish things further:

1. Hematopoiesis Section 

  • Excellent coverage of CD34⁺ progenitors and megakaryocyte regulation via SHP1.
  • If possible, add a brief note on how LAIR1 might affect long-term HSC self-renewal or lineage choice.

2. AML & CLL Discussion 

  • Your deep dive into GM-CSF/Akt/Ca²⁺ inhibition in AML and the CLL prognostic correlations is spot on.

  • To strengthen the immune-escape angle, consider a few sentences on how LAIR1 might interact with PD-1/PD-L1 in the leukemic microenvironment or influence resistance to existing treatments.

3. Soluble LAIR2 & Biomarkers 

  • The NC410 story is compelling. You might mention whether serum LAIR2 levels have been—or could be—examined as a biomarker in leukemia patients.

4. Minor Formatting

  • Double-check that section and figure numbers match their captions.
  • In your conclusion, clarify whether LAIR1-targeting strategies are intended solely for hematologic malignancies or also in combination with solid-tumor therapies.

With these small tweaks, your review will read smoothly and maintain the strong, clinically relevant narrative you’ve built. Congratulations on a much-improved manuscript!

Author Response

Dear Authors,

Thank you for your thorough revision. The new material really brings out LAIR1’s importance in normal and malignant blood cells. A few friendly suggestions to polish things further:

  1. Hematopoiesis Section 
  • Excellent coverage of CD34⁺ progenitors and megakaryocyte regulation via SHP1.
  • If possible, add a brief note on how LAIR1 might affect long-term HSC self-renewal or lineage choice.
  • We replied to this question inserting the following sentences (lines 1090-1094):

It is not defined the relevance of LAIR1 on HSC self-renewal and lineage differentiation. It might be hypothesized that growth factors involved in proliferation and differentiation of HSC into specific lineage can influence differently the level of expression of LAIR1. The study of this possibility should be faced before using targeting of LAIR1 in hematological malignancies.

  1. AML & CLL Discussion 
  • Your deep dive into GM-CSF/Akt/Ca²⁺ inhibition in AML and the CLL prognostic correlations is spot on.
  • To strengthen the immune-escape angle, consider a few sentences on how LAIR1 might interact with PD-1/PD-L1 in the leukemic microenvironment or influence resistance to existing treatments.
  • We replied to this question inserting the following sentences (lines 1155-1163):

This point is essential to define whether the blocking of both PD1 and LAIR1 can be used in combination. In the TME, it is conceivable that PD1 interacts with PDL1 on tumor cells while LAIR1 binds to collagen in the stroma. This would suggest that effector immune cells could be strongly stimulated to respond due to the lack of the tumor- and the stroma-mediated inhibition. However, the biochemical mechanisms activated simultaneously by PD1 and LAIR1 when targeted with blocking antibodies should be studied in detail to tune these two potent immune regulatory checkpoints. On the other hand, the side effects of this blockade should be considered, as PD1 is involved in adaptive immune response and LAIR1 can target the innate arm of the immune system as well.

  1. Soluble LAIR2 & Biomarkers 
  • The NC410 story is compelling. You might mention whether serum LAIR2 levels have been—or could be—examined as a biomarker in leukemia patients.
  • To our knowledge the serum level of LAIR2 has not been analyzed in leukemia.  We mention the possibility of determine LAIR2 levels in leukemia as a biomarker as follows (lines 1172-1175):

The levels of LAIR2 could be evaluated both in hematological and solid tumors to plan the targeting of LAIR1-mediated inhibitory effects. Further, the LAIR2 might be studied as a biomarker of some leukemia therapy response or as a mechanism of resistance.

  1. Minor Formatting
  • Double-check that section and figure numbers match their captions.
  • We checked the caption and the figure. Please note that the table IV appears like a figure but it is a table.
  • In your conclusion, clarify whether LAIR1-targeting strategies are intended solely for hematologic malignancies or also in combination with solid-tumor therapies.
  • We replied to this question, inserting the following sentences (lines 1209-1212):

However, the blocking of the LAIR1-mediated signal in tumor-infiltrating lymphocytes should elicit a strong antitumor immune response of both T and NK cells. This could be the rationale to use targeting of LAIR1 in solid tumor therapy as well.

With these small tweaks, your review will read smoothly and maintain the strong, clinically relevant narrative you’ve built. Congratulations on a much-improved manuscript!
